# HIV-1 Reverse Transcriptase Expression in HPV16-Infected Epidermoid Carcinoma Cells Alters E6 Expression and Cellular Metabolism, and Induces a Hybrid Epithelial/Mesenchymal Cell Phenotype

**DOI:** 10.3390/v16020193

**Published:** 2024-01-26

**Authors:** Alla Zhitkevich, Ekaterina Bayurova, Darya Avdoshina, Natalia Zakirova, Galina Frolova, Sona Chowdhury, Alexander Ivanov, Ilya Gordeychuk, Joel M. Palefsky, Maria Isaguliants

**Affiliations:** 1Chumakov Federal Scientific Center for Research and Development of Immune-and-Biological Products of Russian Academy of Sciences, 119991 Moscow, Russia; bayurova_eo@chumakovs.su (E.B.); avdoshina_dv@chumakovs.su (D.A.); frolova_ga@chumakovs.su (G.F.); gordeychuk_iv@chumakovs.su (I.G.); 2Gamaleya National Research Center for Epidemiology and Microbiology, 123098 Moscow, Russia; aivanov@yandex.ru; 3Centre for Precision Genome Editing and Genetic Technologies for Biomedicine, Engelhardt Institute of Molecular Biology, 119991 Moscow, Russia; nat_zakirova@mail.ru; 4Division of Infectious Diseases, Department of Medicine, University of California, San Francisco, CA 94143, USA; sona.chowdhury@ucsf.edu (S.C.); joel.palefsky@ucsf.edu (J.M.P.); 5Department of Microbiology, Tumor and Cell Biology, Karolinska Institutet, 171 77 Stockholm, Sweden

**Keywords:** HIV-1, reverse transcriptase, HPV16-positive cells, E6*I isoform expression, glycolysis, mitochondrial respiration

## Abstract

The high incidence of epithelial malignancies in HIV-1 infected individuals is associated with co-infection with oncogenic viruses, such as high-risk human papillomaviruses (HR HPVs), mostly HPV16. The molecular mechanisms underlying the HIV-1-associated increase in epithelial malignancies are not fully understood. A collaboration between HIV-1 and HR HPVs in the malignant transformation of epithelial cells has long been anticipated. Here, we delineated the effects of HIV-1 reverse transcriptase on the in vitro and in vivo properties of HPV16-infected cervical cancer cells. A human cervical carcinoma cell line infected with HPV16 (Ca Ski) was made to express HIV-1 reverse transcriptase (RT) by lentiviral transduction. The levels of the mRNA of the E6 isoforms and of the factors characteristic to the epithelial/mesenchymal transition were assessed by real-time RT-PCR. The parameters of glycolysis and mitochondrial respiration were determined using Seahorse technology. RT expressing Ca Ski subclones were assessed for the capacity to form tumors in nude mice. RT expression increased the expression of the E6*I isoform, modulated the expression of *E-CADHERIN* and *VIMENTIN*, indicating the presence of a hybrid epithelial/mesenchymal phenotype, enhanced glycolysis, and inhibited mitochondrial respiration. In addition, the expression of RT induced phenotypic alterations impacting cell motility, clonogenic activity, and the capacity of Ca Ski cells to form tumors in nude mice. These findings suggest that HIV-RT, a multifunctional protein, affects HPV16-induced oncogenesis, which is achieved through modulation of the expression of the E6 oncoprotein. These results highlight a complex interplay between HIV antigens and HPV oncoproteins potentiating the malignant transformation of epithelial cells.

## 1. Introduction

People living with human immunodeficiency virus (HIV-1) (PLWH) are at an increased risk of developing epithelial cell cancer, even with long-term successful antiretroviral therapy [1]. One of the driving mechanisms was suggested to be the direct tumorigenic effects of some HIV-1 proteins, specifically their ability to cause the malignant transformation of epithelial cells [2]. HIV proteins tat, nef, gp120, matrix protein p17, and reverse transcriptase (RT) induce oxidative stress with serious consequences in the form of DNA, protein, and lipid damage, as well as changes in intracellular signaling, and have a direct carcinogenic potential in vivo [3,4,5]. Another potent driver is co-infection with oncogenic viruses, such as the hepatitis B virus (HBV), hepatitis C virus (HCV), Epstein–Barr virus (EBV), or cytomegalovirus (CMV) [6]. Individuals with HIV, especially with acquired immunodeficiency syndrome (AIDS), are at a specifically high risk of developing cancers associated with infection with human papilloma viruses (HPV) [7,8,9,10,11,12]. Infection with high risk HPVs (HR HPVs) is responsible for most cases of cervical and anal cancer, as well as a subset of cancers of the oropharynx, penis, vagina, and vulva [13,14,15,16]. Women living with HIV are six times more likely to develop cervical cancer compared to women without HIV, and an estimated 5% of new cervical cancer cases are attributable to HIV [15]. Cervical cancer in HIV-infected women shows specific features: it occurs at a younger age than in the general population, reveals advanced stages at presentation, develops metastases in unusual locations, demonstrates a poor response to treatment, a higher recurrence rate, and a shorter time interval to death [17]. Anal cancer is also more common in immunocompromised individuals, specifically PLWH [18,19]. Even in HIV patients with CD4 cell counts over 500/μL, the risk of developing anal cancer was shown to be more than 20-fold higher than among the general population [20]. Anal infections with HR HPVs are very common among HIV-infected women and men who have sex with men (MSM). The unadjusted anal cancer incidence rates per 100,000 person years were reported to be 2 for HIV-uninfected men, 46 for other HIV+ men and 131 for HIV + MSM [21,22,23,24].

Cooperation between HIV-1 and HR HPVs, specifically, HPV16 in the malignant transformation of epithelial cells has long been anticipated, with HR HPVs infecting epithelial cells [25,26], and HIV-1 entering these cells by multiple unconventional mechanisms such as transversion of the epithelial lining or “natural pseudotyping” [27,28,29,30]. HIV-1 can also release into extracellular space viral antigens which affect the “innocent” bystander cells [2,31]. This enables an interaction between HIV-1 and HPV viral proteins in HPV-infected epithelial cells. The possibility of molecular interactions between these viruses and/or their antigens leading to an enhanced risk of developing cancer have been independently addressed in several studies. As an example, HIV-1 nef is transported from HIV-1-infected cells to the neighboring target cells via filopodia and/or exosomes [32,33,34] and interacts with the ubiquitin (Ub)-protein ligase E3A (UBE3A/E6AP) complex [35] targeting the tumor suppressor p53 for ubiquitination and degradation, thereby contributing to HPV-induced cervical carcinogenesis [36,37,38]. The effects of HIV-1 and HR HPVs can be independent or bidirectional, potentiating one another.

As of today, the data on the direct interactions between HPV and other HIV-1 proteins is limited. HIV-1 tat was shown to increase the expression of HPV E6 and E7 oncoproteins, enhancing the E6 and E7 mediated oncogenic effects of HPV [39,40,41,42], as well as to increase E2 transcription, which modulated HPV replication [43]. HIV-1 rev was found to increase the expression of HPV L1 [44], a phenomenon mechanistically unclear given the nuclear localization of Rev in HIV-1-infected cells. The same applies to other HIV-1 antigens: the nuclear protein vpr involved in cell cycle regulation [45], the envelope protein gp120, and reverse transcriptase (RT). Vpr was shown to interact with E6 in cervical cancer cells [46]. Both gp120 and RT increase the expression of HPV16 E6 in HPV16 infected immortalized and/or completely transformed epithelial cells, while other HIV-1 proteins, such as capsid protein p24, had no effect [2]. Furthermore, the prolonged interaction of HIV-1 proteins gp120 and tat and cell-free HIV-1 virions with HPV16-immortalized anal, cervical, and oral epithelial cells was found to stimulate EMT and increase the invasiveness of HPV16-infected cells [47,48]. Overall, these data indicate that the presence of HIV-1 and/or its antigens in HPV16-infected neoplastic cells may potentiate HPV-associated tumorigenicity, making HIV-1 infection an integral part of the process of HPV-associated tumorigenesis.

Earlier, we have shown that the expression of HIV-1 RT in murine adenocarcinoma cells induces the production of reactive oxygen species (ROS) and genetic instability; this, in turn, increases the expression of factors promoting the epithelial–mesenchymal transition (EMT). Furthermore, RT-expressing cancer cells acquire an enhanced capacity to grow and metastasize in immunocompetent mice [2,5]. The aim of this study was to evaluate the effect of HIV-RT on HPV16-infected human epithelial cancer cells and to determine whether RT can modulate the phenotypic characteristics, metabolic activity, gene expression patterns, and in vivo tumorigenicity of these cells.

## 2. Materials and Methods

### 2.1. Lentiviral Transduction of Ca Ski Cells and Isolation of Clones Expressing RT_A and GFP Variants

The coding sequence for the consensus HIV-1 RT_A (RT_A) was synthesized by Evrogen (Moscow, Russia). The coding sequence for RT_A was re-cloned from p6HRT_A into the lentiviral vector pRRLSIN.cPPT.PGK (Addgene, Watertown, MA, USA) generating the lentiviral vector pLV-RT_A as described earlier [5]. The production of lentiviral particles was carried out in HEK293T culture cells by transfection of the pLV-RT_A in the presence of a polyethyleneimine (PEI) (MW 25000, Polysciences, Warrington, PA, USA), as described earlier [49]. After 48 h from transfection, cell culture medium containing viral particles was harvested, clarified by low-speed centrifugation, and filtered through a 0.45 μm sterile filter, followed by concentration using Amicon Ultra 100 K centrifuge concentrators (Merck-Millipore, Darmstadt, Germany).

Infectious titers of the lentiviral particles were determined in HeLa cells by quantitative real-time PCR using standard samples of HeLa DNA with a known number of copies of the provirus [50]. Transduction of HPV16-infected human cervical squamous cell carcinoma Ca Ski cells (CRL-1550) was performed with the multiplicity of lentiviral infection (MOI) of 5 and 10 transducing units per cell without the addition of polybrene. Monoclonal populations of Ca Ski derivative clones were generated by limiting dilution in 96-well plates. Four subclones containing RT_A DNA were re-cloned by limiting dilution, generating three derivatives from the first-round subclone B8 (MOI 5; B8B5, B8D5, B8D2) and three from the subclone H6 (MOI 10; H6G11, H6D7, H6F8). The resulting Ca Ski derivatives were cultured in the full RPMI-1640 (Paneco, Moscow, Russia) medium with 100 mg/mL penicillin/streptomycin (Paneco) mix and 12% FBS (HyClone, Logan, UT, USA) (due to reduced re-cloning efficiency and slow growth rate) at 37 °C with 5% CO_2_ and split every 2–3 days.

Control cell lines expressing a fluorescent marker were obtained with the help of the lentiviral vector pRRLSIN.PGK.EGFP (Addgene) encoding GFP under the control of the polyglycerate kinase (PGK) promoter. The transduced cells were cloned into 96-well plates by limiting dilution. One individual subclone was selected based on the presence of a fluorescent signal. The stability of GFP gene expression after two weeks in culture was confirmed using a flow cytometer (Beckman Coulter CytoFLEX, San Jose, CA, USA).

### 2.2. Determination of the Number of Copies of the Provirus in the Genome of the Transduced Lines

A comparative analysis of the number of copies of the lentiviral genome in the DNA of the Ca Ski derivatives was carried out using real-time PCR according to the protocol used to determine the infectious titer [50]. The ∆Ct was defined as the difference between the threshold values of two amplification reactions with primers to the housekeeping gene (beta-actin) and to the 5′-region of the provirus genome (Appendix A). The number of copies of the provirus was determined using a calibration curve using control DNA samples of HeLa cells with a known number of copies of the provirus [50].

### 2.3. Confirmation of RT Production by Western Blotting

Production of RT_A protein was analyzed by Western blot of cell lysates. Lysates of 10^5^ cells of each RT variant were resolved by PAGE and subjected to Western blotting using polyclonal anti-RT rabbit antibodies [51] and anti-ß-actin monoclonal antibodies (AC-15, Invitrogen, Waltham, MA, USA). The lysate of the parental Ca Ski cell line was used as a negative control. Recombinant RT_A obtained as described earlier [5] was loaded onto the gel in fixed amounts from 1 to 27 ng per well and was used as a positive control to build a calibration curve. Using this curve, we assessed the amount of RT in the aliquots of cell lysates loaded onto the gel, and by dividing this value by the number of cells used to prepare the lysate, we determined the levels of RT expression per cell for each derivative clone. Signals from the bands were quantified by ImageJ software (version 9.4.0, http://rsb.info.nih.gov/ij (accessed on 21 July 2021)).

### 2.4. Isolation of Nucleic Acids, Reverse Transcription, and Semiquantitative PCR

The cell culture medium was discarded; cells were detached using 0.05% trypsin with EDTA (Paneco) and centrifuged at 600× *g*. RNA was isolated using the GeneJET RNA Purification Kit (cat. K0731, Thermo Fisher Scientific, Waltham, MA, USA) according to the manufacturer’s instructions and reverse transcription was performed using the MMLV RT kit (Evrogen, Moscow, Russia).

Gene-specific PCRs were performed on a RotorGene 6000 (Qiagen, Hilden, Germany) cycler using primers specific to *RT_A*, *E6FL*, *E6*I*, *E6*II*, *E7*, *NRF2*, *NQO1*, *GCLC*, *A-TUBULIN*, *Y-TUBULIN*, *N-CADHERIN* and *VIMENTIN*, *E-CADHERIN*, *TWIST1*, *SNAI1,* and *SNAI2* genes and SYBR Green (Evrogen) (Appendix A). Levels of mRNAs were measured relative to *GAPDH* or *GUSB* mRNA. Relative gene expression levels were calculated using the ddCt method [52].

### 2.5. Cell Culture and Microscopic Quantitation of Proliferation

Ca Ski and their derivative cells were maintained in RPMI-1640 (Paneco) with 10% FBS (HyClone). The doubling time of derivative clones was estimated as described previously [5]. Briefly, adherent cells were washed with PBS and detached with 0.05% trypsin with EDTA (Paneco). Thereafter, 10 μL of the resulting suspension was transferred to a Gorjaev’s count chamber and cells were counted (×5). Scoring of the proliferation rate was performed by counting cells in 20 fields of the counting grid.

The population doubling time, or the time required for a culture to double in number were calculated using the following formula: DT = T ln2/ln(Xe/Xb), where T is the incubation time in a given time unit; Xe is the cell number at the end of the incubation time; and Xb is the cell number at the beginning of the incubation time (https://www.atcc.org/resources/culture-guides/animal-cell-culture-guide (accessed on 14 June 2021)).

### 2.6. Measurement of Glycolysis and Mitochondrial Respiration

Glycolysis and mitochondrial respiration were assessed using Seahorse technology on a XFe24 analyzer (Agilent Technologies, Santa-Clara, CA, USA) according to the manufacturer’s instructions with a number of modifications [53]. Briefly, 24 h prior to analysis, the cells were seeded into an XF24 Cell Culture Microplate (3 × 10^4^ cells/well) in RPMI-1640 (Paneco) with 10%FBS (HyClone), with four replicates.

For the MitoStress test, 45 min before analysis, the media were changed to DMEM (Gibco, Thermo Fisher Scientific) lacking phenol red dye and bicarbonate and supplemented with 2 mM glutamine. The plate was kept at 37 °C, in a normal atmosphere. To evaluate respiration-linked ATP production, maximum respiratory capacity, and non-mitochondrial respiration, ATP-synthase inhibitor oligomycin, uncoupler FCCP, and a mixture of complex I and III inhibitors rotenone and antimycin were added to final concentrations of 1 µM (Oligomycin), 0.75 µM and 1.5 µM (FCCP), and 1 µM each of rotenone/antimycin. For each condition, three readings were performed at 3 min intervals.

In the case of the GlycoStress test, 30 min prior to analysis, the medium was changed to DMEM lacking phenol red dye, bicarbonate, and glucose, and supplemented with 2 mM glutamine. During the assay 11 mM and 30 mM glycose, 1 µM oligomycin and 50 mM 2-deoxyglucose were added to access basal glycolysis, maximal glycolytic capacity, and non-glycolytic acidification, respectively.

After the analysis, the cells were lysed, and the total protein concentration was measured using the PierceBCA Protein Assay Kit (Thermo Fisher Scientific) according to the manufacturer’s instructions. The MitoStress and GlycoStress test results were normalized to the total protein levels. The raw data were processed using Seahorse Wave Desktop software (Agilent Technologies) and further analyzed using GraphPad Prism version 9.4.0 software (GraphPad Software, La Jolla, CA, USA).

### 2.7. Assessment of the Production of Reactive Oxygen Species (ROS)

The production of ROS was accessed using redox-sensitive fluorescent probes: dihydroethidium (DHE) and 2′,7′-dichlorodihydrofluoresceine diacetate (DCFH2-DA). The cells were stained as described earlier [54]. Stained cells were incubated at room temperature for 30 min in a fresh medium containing 25 μM DCFH2-DA, washed 10 times with 0.5 mL PBS, and resuspended in 200 μL PBS. The fluorescence intensities (FLI) were recorded on a Plate CHAMELEON V reader (Hidex Ltd., Turku, Finland). Specific levels of superoxide anions were measured after treating the cells with 25 μM dihydroethidium (DHE) using the protocol given above for DCFH2-DA. In the case of DHE, FLI were measured with excitation at 510 nm and emission at 590 nm, and in the case of DCFH2-DA, with excitation at 485 nm and emission at 535 nm.

### 2.8. Wound Healing Assay

The effect of RT_A and GFP expression on Ca Ski cell migration was assessed using the wound healing assay [55]. Cells were grown in 96-well plates until a 100% monolayer was achieved. The surface of the well was scratched with a pipette tip. Eight repeats per each cell subclone were washed with PBS followed by medium change and continuously imaged using CELENA^®^ X High Content Imaging System (Anyang-si, Gyeonggi-do, Republic of Korea) at 4× magnification for 24 h. The images were processed using the ImageJ/Fiji^®^ module, quantifying the entire area of the wound, the average width of the wound and the standard deviation of the width in all images. Data reflecting the change in the area values (size of the wound) for each individual repeat during each hour of monitoring were used to build a slope-intercept from y = mx + b, where “m” is the slope (or steepness) of a line, “b” is the y-intercept where the line crosses the *y*-axis, and therefrom calculate the slope parameter by setting y = b/2 (i.e., the point at which the gap is half the original area) and solving for x. The speed of migration (U) was calculated as U = |slope, m|/2×L, where L is the length of the wound [55].

### 2.9. Clonogenic Assay

To analyze the effect of RT on clonogenic activity, cells were plated in 60 mm dishes with 200 cells and incubated under standard conditions, after which single cells formed colonies. The mean size and number of the colonies were measured at 21 days post seeding. Colonies were washed three times with PBS, fixed with 100% methanol, and stained with 0.5% crystal violet. Thereafter, their number and mean size were quantified using ImageJ software (https://imagej.net/ij/download.html (accessed on 21 July 2021)). Each assay was performed in triplicate.

### 2.10. Cell Cycle Analysis

The representation of Ca Ski subclones in different stages of the cell cycle was assessed by flow cytometry as described previously [56]. In brief, cells were cultured in a T25 culture flask, detached, sedimented, and washed twice with PBS. Thereafter, cells were fixed with chilled 70% ethanol, incubated on ice for 30 min, washed again with cold PBS, stained with propidium iodide (PI) dye (25 µg/mL), and treated with the solution of RNase A (50 µg/mL) followed by incubation at room temperature for 30 min. Finally, the cell cycle distribution of the cells was carried out by using a flow cytometer (Beckman Coulter CytoFLEX, San Jose, CA, USA). The distribution of cells in G1/G0, S, and G2/M areas was assessed using FlowJo software (https://www.flowjo.com (accessed on 29 November 2021)).

### 2.11. Assessment of Tumorigenicity of Ca Ski Derivative Clones

The capacity of the derivative cell lines to form tumors was tested by ectopic implantation into 8-week-old female Nu/J mice. The animal experiments were performed in the animal facilities of the NF Gamaleya Research Center of Epidemiology and Microbiology (GRCEM, Moscow, Russia), Ministry of Health of the Russian Federation (Moscow, Russia). The experiments were carried out in accordance with the bioethical principles adopted by the European Convention for the Protection of Vertebrate Animals used for Experimental and Other Scientific Purposes (Strasbourg, France, 1986). The experimental procedures were approved by the ethical committee of the Gamaleya National Research Center for Epidemiology and Microbiology (protocol N10, 14 March 2017 prolonged to protocol N39, 24 March 2023). Six-week-old 17 g Nu/J mice from the nursery of the Center for Collective Use “SPF-vivarium” of the Federal Research Center Institute of Cytology and Genetics, Siberian Branch of Russian Academy of Sciences (Novosibirsk, Russia) were housed under a 12 h/12 h light–dark cycle with ad libitum access to water and food and used for experiments after two weeks of adaptation.

Prior to injection, Ca Ski and derivative clones grown in the RPMI-1640 (Paneco) with 10% FBS (HyClone) were detached, sedimented, washed with serum-free medium, and then stained for viability with trypan blue dye (Life Technologies, Carlsbad, CA, USA). They were then counted in a Gorjaev count chamber and aliquoted 10^5^ in 50 µL and 10^6^ in 100 µL of RPMI-1640 in sterile tubes. Aliquots were injected subcutaneously into two sites, to the right and to the left of the base of the tail with a 25 G needle mounted on an insulin syringe (B Braun, Melsungen, Germany) (for injection scheme, see Figure 1).

Tumor size was assessed by morphometric measurements obtained at regular intervals on days 3, 5, 8, 11, 15, 18, 22, 25, and 29 after injection using calipers. Tumor volume was calculated using the standard formula for xenograft volume [57]: V = L × W^2^/2, where V is the tumor volume, L is the length (the longest dimension), W is the width (the distance perpendicular to and in same plane as the length). The humane end point of the experiment was considered when one of the parameters (length and/or width) of the tumor reached 1 cm. At the end of the experiment, 29 days after implantation, the mice were anesthetized with Zoletil 100 (Virbac, Carros, France) and humanely euthanized by cervical dislocation. The tumors and spleens were surgically excised and weighed.

### 2.12. Statistical Analysis

Data were analyzed using nonparametrical and parametrical statistics (GraphPad Prism 9 Software, San Diego, CA, USA). Continuous but not normally distributed variables, such as doubling time, cell cycle, count and size of colonies, tumor volume, and tumor weight were compared in groups using Dunn’s test for pairwise multiple comparisons of the ranked data as the post hoc test following the Kruskal–Wallis test and pairwise using the Mann–Whitney U test with Bonferroni correction. Normally distributed variables were analyzed using the Unpaired *t* test with Bonferroni correction. Correlation analysis was performed using the Spearman Rank Correlation test. For all analyses, *p* < 0.05 was considered significant.

## 3. Results

### 3.1. Lentivirally Transduced Ca Ski Cells Express HIV-1 Reverse Transcriptase

For the human cervical squamous carcinoma cell line Ca Ski with an integrated HPV16 genome (CRL-1550; [58]), the supplementation of HIV-1 RT variants into the culture medium led to an increase in the expression of HPV16 E6 [2]. Here, we aimed to assess whether RT has a systemic effect on Ca Ski cells. Cells were transduced with lentiviral particles expressing HIV-1 RT under the control of the human phosphoglycerate kinase gene (hPGK) promoter. After two rounds of single cell cloning, we obtained a panel of subclones, three derived from subclone B8 with one copy of the RT_A coding sequence per cell genome (B8B5, B8D5, B8D2; Table 1), and three derived from subclone H6 with six copies of the RT_A coding sequence per cell genome (H6G11, H6D7, H6F8; Table 1). The expression of the RT mRNA was confirmed by RT PCR, and of the RT_A protein, by Western blotting (Table 1; Figure 2A,B). All RT_A Ca Ski subclones produced a protein with the expected molecular mass of 66 kDa, specifically recognized by anti-RT antibodies [59] (Figure 2A). Subclones with six RT_A DNA inserts were characterized by a higher level of RT_A production compared to subclones with one insert (Table 1; Figure 2B). The level of RT_A expression per cell (determined based on the relative levels of RT_A and a calibration curve built using recombinant RT [5]) varied from 20 to 55 fg per cell (Table 1).

An effective strategy for mitigating the nonspecific effects of viral transduction on cell line properties is to generate control cells expressing unrelated proteins. This enables the assessment of both the effects of lentiviral transduction per se, and of the metabolic burden due to the overexpression of exogenic protein [60,61]. Here, we obtained control cells by the transduction of Ca Ski with a lentivirus carrying a GFP coding sequence under the control of the PGK promoter, as was done for RT_A. The respective Ca Ski GFP G9 subclone carried six copies of the GFP gene per genome. The expression of GFP was confirmed by flow cytometry (Appendix A).

### 3.2. RT Increases the Expression of the HPV16 E6*I Isoform

Of the cell line properties that RT_A expression can modulate, we first evaluated the levels of the expression of the mRNA of full-length E6 (*E6FL*), its constitutive isoforms *E6*I* and *E6*II* [62,63], and *E7*. The expression levels of *E6FL* (full-length), *E6*II*, and *E7* were increased in single subclones, but the effect was not associated with the expression of RT_A and could have been due to lentiviral transduction (Appendix A). *E6*I* isoform expression was increased in Ca Ski GFP control cells as compared to Ca Ski, with the difference attributable to the effect of lentiviral transduction (Figure 3A). Moreover, all RT-expressing subclones, with one as well as with six RT_A DNA inserts, demonstrated the increased expression of the *E6*I* isoform over that observed in the Ca Ski GFP control (Figure 3A,B). Interestingly, we observed a decrease in *E6*I* expression in clones with six copies of the RT_A DNA insert compared to one copy, suggesting that the overexpression of RT_A was mitigating a nonspecific effect of lentiviral transduction (Figure 3B). The effect was not related to the level of RT_A expression, as there was no difference in the levels of *E6*I* in Ca Ski subclones with a high and a low level of RT_A expression (Figure 3B).

Thus, the constitutive expression of HIV-1 RT in Ca Ski cells increased the level of expression of *E6*I* mRNA, but had little effect on the level of the transcripts of *E6*II* isoform, or of the full length *E6*, or *E7*. Since the E6*I isoform plays a crucial role in HPV16-driven carcinogenesis [64], we have launched a study of the features of Ca Ski subclones associated with their tumorigenicity in vitro and in vivo.

### 3.3. HIV-1 RT Expression Increases the Extracellular Acidification Rate and Decreases the Oxygen Consumption Rate of Ca Ski Cells

We sought to find out whether RT, when expressed in Ca Ski, caused changes in their metabolism, specifically, in glycolysis and mitochondrial respiration. Changes in glycolysis were quantified by assessing the extracellular acidification rate (ECAR) of the culture medium, and changes in substrate utilization in mitochondrial respiration were quantified by measuring the oxygen consumption rate (OCR).

#### 3.3.1. HIV-1 RT Expression in Ca Ski Cells Increases Glycolysis

During glycolysis, glucose is converted to pyruvate, which is transported into mitochondria by subsequent incorporation into the Krebs cycle or secreted in the form of lactate. Enhanced aerobic glycolysis constitutes an additional source of energy for tumor proliferation and metastasis [65]. The efficiency of glycolysis was evaluated using the Seahorse analyzer, which allows for the evaluation of the levels of the acidification of the culture medium using the GlycoStress test method (Figure 4A). Firstly, acidification was assessed in the absence of glucose (the level of non-glycolytic acidification, which is caused by cell processes other than glycolysis; Figure 4A), and thereafter, in the presence of glucose in low- (11 mM; Figure 4A,B) and high-glucose DMEM medium (30 mM, Figure 4A,C). Further, we assessed acidification in cells treated with the mitochondrial ATPase inhibitor oligomycin, which activates glycolysis to the maximum possible level to compensate for abrupted ATP synthesis [66] (Figure 4A,D), and lastly, in the cells treated with the inhibitor of the first stage of glycolysis, 2-deoxy-D-glucose (2-DG) (Figure 4A). The glycolytic reserve was calculated as the difference between the maximum glycolytic capacity and the level of glycolysis at 30 mM of glucose (Figure 4E).

The Ca Ski subclones with six RT_A gene inserts demonstrated increased acidification at low and high glucose levels, reflecting an increase in the basal and maximal glycolysis, respectively (Figure 4B,C), and an increased maximum glycolytic capacity (Figure 4D). No changes were observed in the Ca Ski subclone expressing GFP (Figure 4B–D). Hence, the above changes were not associated with either lentiviral transduction, or metabolic burden on the cells due to the (over) expression of a foreign protein. Thus, the constitutive expression of RT_A in Ca Ski cells caused an increase in the intensity of glycolysis under aerobic conditions. A decrease in the glycolytic reserve was observed in both the GFP and six RT-expressing subclones and could have been due to lentiviral transduction (Figure 4E).

#### 3.3.2. HIV-1 RT Expression in Ca Ski Cells Suppresses Mitochondrial Respiration

Mitochondrial respiration shows the activity of the oxidative phosphorylation system and the related processes of the Krebs cycle, which provides the respiratory system with substrates.

The influence of HIV-1 RT_A on the respiratory activity of mitochondria was assessed on a Seahorse analyzer using the MitoStress method (Figure 5A). Firstly, we determined the basic level of respiration (Figure 5A,B), then the decrease in respiratory activity after treatment with oligomycin which inhibits ATP synthase; the data obtained reflected the level of ATP synthesis by the oxidative phosphorylation system (Figure 5A,C). Thereafter, the respiration was induced to the maximum possible level by the addition of the proton transport uncoupler carbonyl cyanide-4-trifluoromethoxyphenylhydrazone (FCCP) (Figure 5A,D). The difference between the levels of basal and maximum respiration defined the “spare capacity” [67] (Figure 5E). At the final stage, we assessed the residual level of respiration (non-mitochondrial respiration) by adding the inhibitors of mitochondrial complexes I and III, rotenone and antimycin (Figure 5A,F). The difference between the maximum level of oxygen uptake and the level after the addition of oligomycin reflected the intensity of proton leakage (Appendix A).

The Ca Ski GFP subclone demonstrated a decrease in the basal respiration, ATP production, maximal respiration with low FCCP, and the spare capacity compared to Ca Ski (Figure 5B–D,F), all of which are phenomena attributable to lentiviral transduction. For maximal respiration with high FCCP, there were no statistical differences due to the non-normal distribution of the data (Figure 5E). As most of the mitochondrial respiration parameters in Ca Ski subclones with both one and six RT_A gene inserts (1 RT_A and 6 RT_A) did not differ from each other, these subgroups were pooled and compared to the parameters of Ca Ski GFP. Overall, RT_A expressing Ca Ski subclones had decreased levels of ATP production, maximal respiration with high FCCP, and the spare capacity as compared to Ca Ski GFP (Figure 5C,E,F). The level of proton leak did not differ (Appendix A). Thus, on the background of the lentiviral transduction-associated decrease in mitochondrial respiration parameters, the expression of RT_A has further suppressed mitochondrial respiration beyond the levels observed in Ca Ski cells and in the GFP control.

### 3.4. The Expression of HIV-1 RT Did Not Lead to Changes in the Cytoskeleton

Changes in mitochondrial homeostasis can be due to the changes in the cell cytoskeleton. To determine if this was the case, we tested if the expression of RT affects the structure of microtubule (MT) by assessing the expression of A-tubulin, and whether RT modulates the MT-supported mitochondrial network by assessing the expression of Y-tubulin. No differences were observed between the RT_A and GFP expressing Ca Ski subclones in the levels of the mRNA of either *A-TUBULIN* or *Y-TUBULIN*. The single RT_A subclones and the GFP subclone demonstrated a tendency for decreased levels of *Y-TUBULIN* mRNA, but it did not reach a level of significance (Appendix A). These data imply that the observed changes in the glycolysis and mitochondrial respiration of RT_A-expressing Ca Ski subclones were not due to RT_A induced rearrangements of the cytoskeleton.

### 3.5. The Expression of HIV-1 RT in Epithelial Ca Ski Cells Does Not Induce Oxidative Stress

Switching to a more glycolytic metabolism is associated with a decrease in ROS [68]; while *E6*I* expression increases ROS in HPV16-infected cells [69]. To test what would be the cumulative effect of *E6*I* and RT expression, we assessed the levels of ROS in Ca Ski and in Ca Ski subclones. The levels of ROS production in Ca Ski derivatives were assessed using intracellular fluorescent probes: 2′,7′-dichlorodihydrofluoresceine diacetate (DCFH2-DA) and dihydroethidium (DHE). DCFH2-DA reacts to various types of ROS, including H_2_O_2_, O_2_^•−^, and hydroxyl radicals, yielding a fluorescent product that allows for the assessment of the general redox status of a cell, whereas DHE senses only superoxide (O_2_^•−^) [70,71]. 

We could not use the GFP subclone as a control due to the overlap of the fluorescence emission spectra for DCFH2-DA and GFP. A significant decrease in the overall level of ROS production, compared to that in the parental Ca Ski cells, was observed in four out of six RT_A expressing subclones except for B8D2 and H6D7 (Table 1) (Figure 6A). At the same time, staining with DHE demonstrated no change in the levels of production of superoxide in either in RT_A or control GFP-expressing Ca Ski cells (Figure 6B). Thus, the expression of RT_A in Ca Ski cells caused a reduction in the production of ROS (H_2_O_2_, hydroxyl radicals), but not of superoxide O_2_^•−^. Due to the inability to use GFP subclone as the assay control, we could not distinguish whether this effect was due to lentiviral transduction as such or to the (over) expression of a foreign protein or, specifically, to (over) expression of RT_A.

To circumvent the limitation of the assays with fluorescent probes, we turned to the assessment of the expression of the transcription factors involved in cellular redox balance: nuclear factor erythroid 2–related factor 2 (Nrf2); enzymes performing glutathione biosynthesis, namely glutamate-cystein ligase (GCLC); and the enzyme of Phase II of the oxidative stress response, NAD(P)H quinone oxidoreductase 1 (NQO1). The levels of the mRNA of *NRF2*, *GCLC,* and *NQO1* in the RT-expressing Ca Ski subclones were compared to their levels in the GFP control and in the parental Ca Ski cells. For all subclones, we observed a significant decrease in the expression of *NRF2* and *NQO1* mRNA, but a decrease in the levels of *GCLC* mRNA did not reach the level of significance (Figure 7A–C). No differences were observed between the RT_A and GFP expressing subclones, indicating that the effect was due to lentivirus transduction.

Thus, in contrast to our observations for RT_A expressing mammary gland adenocarcinoma 4T1 cells [72], the stable expression of HIV-1 RT in Ca Ski cells did not increase the production of ROS, or the expression of transcription factor *NRF2* associated with oxidative stress, or of the enzymes of Phase II protection from oxidative stress, i.e., had no effect on the redox balance of Ca Ski cells.

### 3.6. Lentivirus-Transduced Ca Ski Express Reduced Levels of Factors Associated with Epithelial–Mesenchymal Transition

As we have shown that the expression of HIV-1 RT causes a metabolic shift towards glycolysis, we next determined whether RT_A expression modulates the epithelial-to-mesenchymal transition (EMT), since EMT is tightly linked to the metabolism of glucose. To normalize the expression level, we used the glyceraldehyde 3-phosphate dehydrogenase (*GAPDH*) gene, which is one of the most commonly used housekeeping genes. The control GFP Ca Ski subclone demonstrated an increased expression of the mRNA of the factors associated with EMT, namely *VIMENTIN* and *N-CADHERIN*, the two markers of the mesenchymal state which promote the acquisition of spindle-shaped mesenchymal cell morphology [73], while the levels of *E-CADHERIN* mRNA were unchanged compared to the parental cells (Appendix A). In contrast to Ca Ski GFP, Ca Ski RT_A demonstrated an increase in the expression of *E-CADHERIN* mRNA (*p* < 0.05 for one RT_A, and *p* < 0.1 for 6 RT_A subclones; Appendix A), manifesting the loss of the typical heterogeneous morphology of epithelial cells. At the same time, Ca Ski RT_A subclones did not differ from the GFP subclone in the levels of *N-CADHERIN* and demonstrated decreased levels of *VIMENTIN* mRNA (Appendix A). The expression of *TWIST1* did not differ from that in the GFP control (Appendix A). The expression of *SNAI1* and *SNAI2* in Ca Ski RT_A subclones was reduced compared to the Ca Ski GFP control only in single subclones (Appendix A).

However, *GAPDH* is known for its function in cellular metabolism [74], hence, it may not constitute an optimal housekeeping gene for cells demonstrating alterations in glycolysis. Hence, we have also assessed the expression level of EMT factors in relation to the *GUSB* gene. The expression of *VIMENTIN* demonstrated the same pattern as the one observed using *GAPDH* (Figure 8B). At the same time, in the GFP control cells, we observed a decrease in the expression of *E-CADHERIN,* while the expression of *N-CADHERIN* remained unchanged compared to the parental cells. When compared to the GFP control, Ca Ski 6 RT_A subclones demonstrated an increase in the expression of the mRNA of both *E-CADHERIN* and *N-CADHERIN,* as well as a decrease in the expression of *VIMENTIN*, altogether indicating the acquisition of the hybrid epithelial/mesenchymal (E/M) phenotype by these cells (Figure 8A–C). In all transduced subclones, the expression of the transcription factors *TWIST1, SNAI1,* and *SNAI2* did not differ from that in the parental cells (Appendix A).

### 3.7. The Effects of HIV-1 RT Expression on the Phenotypical Properties of Ca Ski Cells—Motility and Clonogenic Activity

We next assessed whether the expression of RT_A had any effect on the phenotypic characteristics of Ca Ski cells that shape the metastatic activity of tumor cells, such as cell cycle progression, doubling time, and cell migration rate. Collective cell migration is the coordinated movement of a group of cells that maintain intracellular connections and is crucial to various biological processes including embryo development, the immune response, and cancer metastasis [75]. Another important criterion for the malignancy of tumor cells is a decrease in the need for growth factors and cytokines secreted by microenvironmental cells, which can be revealed by the analysis of clonogenic activity.

#### 3.7.1. Lentiviral Transduction Causes an Increase in Cell Doubling Time, but Does Not Affect Cell Cycle Progression

Five out of six Ca Ski RT_A subclones as well as Ca Ski GFP had an increased doubling time compared to the control Ca Ski cells over a month of consecutive observations (Kruskal—Wallis, *p* < 0.05, *n* = 12) (Table 2, Appendix A), which remained insignificant for the B8D5 subclone (Table 1). While the Ca Ski GFP cells demonstrated a decrease in the percentage of cells in the G1/G0 phase of the cell cycle as compared to the parental cells (Table 2, Appendix A), the RT_A subclones showed no difference from the parental cells in the distribution of cells in all phases of the cell cycle (Table 2, Appendix A). Thus, the expression of RT_A had no effect on cell cycle progression but might have served to overcome the G1/G0 stop caused by the lentiviral transduction and/or (over) expression of the foreign protein.

#### 3.7.2. The Overexpression of HIV-1 RT Reverses the Inhibitory Effect of Lentiviral Transduction on Cell Mobility

The effect of RT_A expression on Ca Ski cell motility was assessed using a wound healing assay.

Lentiviral transduction decreased cell motility for the control Ca Ski GFP subclone (Appendix A). The expression of RT_A in Ca Ski one copy subclones (1 RT_A) had a discrepant effect on cell motility: the B8D5 and B8D2 subclones showed increased cell motility compared to the GFP control, whereas the motility of B8B5 did not differ from that of the GFP control (for a description of the clones, see Table 1). Ca Ski and RT_A subclones with six inserts (6 RT_A) demonstrated a partial restoration of cell motility: for all three subclones, the motility was significantly higher than that of the GFP subclone (Table 2, Appendix A). Thus, in Ca Ski cells, the expression of RT_A partially compensated for a decrease in cell motility caused by lentiviral transduction and/or the overexpression of the foreign protein. However, the phenotypic parameters, cell doubling time, the distribution of cells by the phases of the cell cycle, and the migration rate were not correlated with each other or to the levels of the RT_A mRNA or protein production by the subclones (Appendix A).

#### 3.7.3. The Loss of Clonogenic Activity after Retroviral Transduction Is Compensated in Ca Ski Cells Expressing High Levels of HIV-1 RT

Next, we performed a clonogenic analysis of Ca Ski subclones, as the formation of clones characterizes the ability of cancer cells to initiate the formation of tumors. By day 21 following lentiviral transduction, the control subclone Ca Ski GFP demonstrated a significant decrease in both colony counts (Figure 9A) and the mean size of the colonies (Figure 9B) as compared to the parental Ca Ski cells. The Ca Ski 1 RT_A also demonstrated a decrease in the number of colonies compared to the GFP control. However, the Ca Ski 6 RT_A were indistinguishable from the parental Ca Ski cells (Figure 9A). Also, the Ca Ski 1 RT_A subclones, as the GFP control, had a decreased colony size while the Ca Ski 6 RT_A subclones were indistinguishable in size from the parental cells (Figure 9B). Thus, while lentiviral transduction suppressed clonogenic activity, significantly decreasing the number and size of the colonies, the (over) expression of RT_A reversed these effects, restoring both colony counts and the mean size of the colonies to the levels similar to those of the parental Ca Ski cells.

### 3.8. The Relationship between the Levels of mRNA and the Protein Production of RT_A and E6*I mRNA and the Properties of Ca Ski Subclones

Finally, we asked whether the in vitro properties of the Ca Ski subclones, their metabolic and phenotypic characteristics, are correlated to the expression of RT_A and/or the E6*I isoform. The results of the series of linear correlation tests of these parameters with the expression levels of RT_A and *E6*I* are presented in Table 3.

RT_A was found to have an effect that was independent from the effect of HPV16 E6 on the metabolism of Ca Ski cells by increasing glycolysis (ECAR at low and high glucose). The latter was specifically sensitive to RT_A at low levels of glucose. The expression of *E6*I* had a positive effect, but it did not reach statistical significance (Table 3). At the same time, RT_A inhibited mitochondrial respiration (ATP production and maximal respiration). In contrast to glycolysis, for these parameters, there was a negative association with *E6*I* expression as well which was highly significant (*p* < 0.01), indicating that the inhibition of mitochondrial respiration was determined by both the levels of RT_A and the levels of *E6*I*. Since the significance (by R value) was higher for RT_A, one can assume that RT_A may have played a leading role by inducing *E6*I* expression. Further, we found that the production of ROS correlated with the levels of *E6*I* mRNA, but not with the levels of RT_A (Table 3). Also, we found a correlation between the expression of *N-CADHERIN,* the levels of RT_A mRNA, and RT_A protein (Table 3).

We further analyzed whether the expression of RT_A as mRNA and as a protein and the expression of E6*I mRNA had any effect on the phenotypic characteristics of Ca Ski subclones and found a correlation of the colony forming capacity of the cells with the levels of E6*I mRNA, but not with the levels of the expression of RT_A (Table 3). Specifically, the expression of E6*I mRNA interfered with colony formation, reducing both the number of colonies and their size (Table 3). There was no quantitative dependence of these processes on the amount of RT stimulating *E6*I* expression (despite the RT-mediated increase in the production of the *E6*I* isoform).

### 3.9. The Overexpression of RT Rescues the Tumorigenic Activity of Ca Ski Subclones Affected by Retroviral Transduction

The effect of HIV-1 RT expression on the tumorigenic activity of Ca Ski was assessed by its well-characterized ability to form tumors in the immunosuppressed mice [76]. The experiment was performed on two Ca Ski subclones, one carrying one, and the other carrying six copies of RT_A DNA (B8B5 and H6G11, respectively; Table 1). The parental Ca Ski and its GFP derivative were used as controls. Cells were implanted subcutaneously at a dose of 10^5^ and 10^6^ in two injection sites [77].

Tumor growth in the Ca Ski control group implanted at 10^6^ cells was observed starting from day 11, in the Ca Ski GFP group, from day 15, and for the 6 RT_A subclone H6G11 from day 22 post-injection (Figure 10A). Subclone B8B5 produced no tumors by the experimental endpoint, which was consistent with its reduced clonogenic activity in vitro (Appendix A). By the experimental endpoint on day 29, a significant difference in the tumor growth rate was noted only between the parental Ca Ski and the Ca Ski GFP control, with the latter growing much slower than the parental cells (*p* = 0.0195; Figure 10D). The H6G11 subclone did not differ from the parental Ca Ski cells (*p* > 0.1; Figure 10D) in its ability to form solid tumors in immunosuppressed mice, although not all injections resulted in palpable tumors. The late start of tumor growth and late entry into the exponential growth phase did not allow us to evaluate the full effect of the restoration for the RT-expressing subclone of the tumorigenicity of Ca Ski cells affected by lentivirus transduction since the experiment had to be terminated when the first tumor that formed in the mice in the experiment reached 1 mm^3^ in size (observed in the control Ca Ski group). The implantation of 10^5^ cells did not give tumors by day 29 post-injection for any of the subclones.

At the experimental endpoint, the mice were euthanized and the tumors were excised and weighed. A significant difference was observed between the parental Ca Ski and Ca Ski GFP controls, as the latter were much smaller. No difference was observed in the volumes of the tumors formed by the Ca Ski and Ca Ski 6 RT_A subclones (Figure 10E), which was consistent with the growth kinetics assessment by day 29 (Figure 10D,E).

In terms of the ability to form solid tumors in immunosuppressed mice, the 6 RT_A Ca Ski subclone did not differ from the parental cells (*p* > 0.1), while the expression of GFP led to a significant reduction in tumorigenicity. This indicated that although lentiviral transduction and the further synthesis by tumor cells of a non-oncogenic exogenous protein such as GFP decreases their tumorigenicity, the overexpression of HIV reverse transcriptase restores the compromised tumorigenic activity to the levels observed in the parental Ca Ski cells.

## 4. Discussion

The overall objective of this study was to understand the mechanism by which HIV-1 contributes to malignant transformation of the epithelial cells, particularly in the context of its interaction with other oncoviruses such as HPV16. HPV16 is a major contributor to non-AIDS related cancers in HIV-1 infected individuals, with its oncoproteins E6 and E7 playing a pivotal role in carcinogenesis. It is believed that possible molecular interactions between these viruses and/or their antigens lead to an enhanced risk of developing cancer [7,8,9,10,11,12]. However, there is a paucity of data regarding direct interactions between HPV and other HIV-1 proteins. We hypothesized that the HIV-1 protein reverse transcriptase (RT), a multifunctional enzyme, which generates proviral HIV-1 DNA from an RNA template and has three sequential biochemical activities: RNA-dependent DNA polymerase activity, ribonuclease H (RNase H), and DNA-dependent DNA polymerase activity [78], is associated with the malignant transformation of epithelial cells either alone or in collaboration with HPV16 oncoproteins. Previous findings suggest that RT can induce oxidative stress and can be secreted into the extracellular space, potentially influencing neighboring cells [2]. In addition, the expression of HIV-1 RT leads to an increase in mitochondrial respiration and the level of ATP synthesis in cancer cells, which is associated with the restoration of the mitochondrial network [72]. The induction of E6 expression is a property shared by a broad range of reverse transcriptase (RT) variants. Additionally, the indicates that RT, when added extracellularly, has the ability to enter into the cell [2]. Our current data reveal that the expression of HIV-1 RT in HPV16-infected epithelial cells confers a certain adaptability or plasticity, allowing them to maintain key properties, i.e., overcoming the effects of lentiviral transduction. Moreover, these co-infected transformed cells acquire specific features that are associated with increased tumorigenicity.

We have selected a model cell line—Ca Ski cells (HPV-16 infected human cervical squamous cell carcinoma, [58]) to determine whether HIV-1 RT can change their properties. Ca Ski cells were transduced with the lentivirus encoding consensus RT of the HIV-1 clade A FSU_A strain (RT_A) [5]. We generated subclones with one and six genomic inserts of the RT_A coding sequence expressing from 20 to 55 fg protein per cell.

One HIV-1 virion contains approximately 50 molecules of RT [79] as a complex of a 66-kDa polypeptide and its 51-kDa proteolytic digestion product that lacks the carboxy-terminal sequences present in the 66-kDa form [80]. For HIV-1, a single productively-infected CD4+ T cell should carry a viral load of approximately 500 virions [81]. Another estimation provides an average of 3900 (range 3162–5011) viral RNA copies per infected cell, with two RNA molecules per virion, this accounts for approximately 2000 virions per cell [82], i.e., four times more. After recalculation from kDa into fg, the total amount of RT in CD4 + T cells could be estimated as 2.75 fg of the p66 and 2.12 fg of the p51 polypeptide, with a total of 5 fg per infected cell based on the first estimation, and 20 fg per cell based on the second estimation. Thus, Ca Ski subclones produced an adequate amount of HIV-1 RT corresponding to protein levels in a natural infection (although in the epithelial cells, not in CD4+ T lymphocytes).

Lentiviral infection has been reported to change the properties of transduced cells [58,59]. Furthermore, the (over) expression of foreign proteins overloads protein synthesis machinery and can grossly affect the properties of the cells. An effective strategy for mitigating the nonspecific effects of viral transduction on cell line properties is to generate control cells expressing unrelated proteins. To this end, we created a control Ca Ski cell line transduced by a lentivirus encoding GFP, with a multiplicity of infection similar to the highest one used to generate Ca Ski RT_A subclones. All of the in vitro and in vivo properties of the Ca Ski RT_A subclones were compared to the Ca Ski GFP control.

First, we assessed the effect of HIV-1 RT on the expression of HPV16 oncoproteins, determining the level of their transcription in Ca Ski cells. A bicistronic pre-mRNA encodes HPV16 E6 and E7 oncoproteins and alternatively spliced transcripts encoding E6*I and E6*II [83]. RT had no effect on the expression of the mRNA of *E7*, full-length *E6*, or the *E6*II* isoform, but increased the expression of the *E6*I* isoform. This effect was only partly due to lentiviral transduction, as it was also observed in the Ca Ski GFP subclone, but at a significantly lower level. Interestingly, the phenomenon was observed in all RT-expressing subclones, with both one and six RT_A DNA inserts and was not related to the level of RT_A expression (as there was no difference in the levels of *E6*I* in the subclones expressing high and low levels of RT_A). The latter indicated that RT_A was not a direct mediator of *E6*I* expression, but rather an inducer of the expression in a concentration-independent manner.

E6/E7 splicing is regulated by the interaction of cis-acting elements, including branch point sequences (BPSs) and splicing silencers, as well as trans-acting factors. The sequential nucleotides AACAAAC (for HPV16) located in the E6 coding region upstream of the 3′ splice site of SA409 were identified as BPSs that are closely associated with E6*I splicing efficiency and further influence E7 production [84]. A critical point mutation can interrupt the binding activity of BPSs to trans-acting factors, causing inefficient splicing for the production of the E6*I protein, but promoting the expression of significant amounts of E6*II. Thus, it is possible that the RT could tip the balance of E6/E7 expression toward E6*I by promoting binding of BPS to trans-activating factors. 

We next investigated the effects of HIV-1 RT on cell metabolism. The alteration of glucose metabolism is a basic property of cancer cells, as are increases in glucose consumption and glycolytic rates as well as “aerobic glycolysis” or the Warburg effect in the presence of oxygen. Oncoproteins E6 and E7 favor the Warburg effect through an increase in the activity of glycolytic enzymes, as well as the inhibition of the Krebs cycle and the respiratory chain [85]. With this, HPV16-infected Ca Ski cells employ the metabolic pathway of aerobic glycolysis. Furthermore, the artificial overexpression of HPV16 E6/E7 in Ca Ski cells significantly upregulates the glycolysis pathway [86]. One can expect that any factor leading to an increase in E6 expression would tilt these cells further into the glycolytic pathway. Indeed, the constitutive expression of RT_A in Ca Ski cells induced an increase in the levels of E6*I isoform expression and led to an increase in the intensity of glycolysis under aerobic conditions. Ca Ski subclones with six RT_A gene inserts demonstrated an increase in basal and maximal glycolysis and increased the maximum glycolytic capacity, while no changes in the glycolytic pathway were observed in the Ca Ski GFP subclone. At the same time, the expression of RT_A suppressed mitochondrial respiration as compared to the GFP control. An increase (no reduction in our case) in proton leak and a decrease in basal or maximal respiration are indicators of mitochondrial dysfunction [87]. Linear correlation analysis demonstrated a direct correlation of ECAR at low and high glucose. We also noted an inverse correlation of the parameters of mitochondrial respiration with the levels of RT_A mRNA and of RT_A protein, as well as with the levels of E6*I mRNA, although the significance for the latter was lower (had lower R values). This pointed towards the unidirectional effects of HIV-1 RT and E6*I, with RT as a primary factor modifying cell metabolism by altering the amounts of the E6*I isoform. Expressed in Ca Ski cells, HIV-1 RT changed their metabolism by enhancing glycolysis and inhibiting mitochondrial respiration, with the suppressive effect possibly mediated through the modulation of the levels of the E6*I isoform. It was previously shown that another HIV-1 protein, the trans-activator of transcription (Tat) also induces a decreased maximal respiration and reduces spare respiratory capacity in Lund human mesencephalic (LUHMES) cells [88]. In contrast, head and neck squamous cell carcinoma (HNSCC) cells positive for human papillomavirus were found to favor mitochondrial metabolism mediated by HPV16 and HPV18 E6 over glucose metabolism [89]. Differences in the effects observed in Ca Ski RT_A subclones and HNSCC could have been due to HIV-1 RT interference.

The E6-independent effect of HIV-1 RT must also be considered. We have earlier shown RT to be able to alter the metabolism of cancer cells. Interestingly, in adenocarcinoma cells characterized by the high levels of ROS, HIV-1 RT enhanced mitochondrial respiration (OXPHOS), but not glycolysis [72]. The direction of metabolic changes is tightly linked to the redox status of the cell. ROS are involved in glycolysis (dys)regulation. ROS inhibit multiple glycolytic enzymes, including glyceraldehyde 3-phosphate dehydrogenase, pyruvate kinase M2, and phosphofructokinase-1 [90]. In contrast, a shift to glycolysis inhibits the production of ROS. Glycolytic inhibition consistently promotes flux into the oxidative arm (of the pentose phosphate pathway to generate NADPH) [91]. The enhancement of OXPHOS in the background of high ROS reflected this process with HIV-1 RT adapting the adenocarcinoma cells to the high-ROS environment by shifting their metabolism towards mitochondrial respiration.

To determine if this was the case for Ca Ski subclones, we assessed their overall levels of ROS using sensor dyes, DCFH2-DA sensing H_2_O_2_, hydroxyl radical, and superoxide O_2_^•−^, and DHE sensing only superoxide O_2_^•−^. The majority of Ca Ski subclones demonstrated decreased levels of total ROS, while the levels of superoxide production remained unchanged compared to the parental cells. We could not delineate a specific effect of RT_A on the total ROS due to the overlap of the signals of DCFH2-DA and GFP. To compensate for this, we assessed the levels of transcription factors induced in response to the oxidative stress: Nrf2; the enzymes of glutathione biosynthesis, GCLC; and of the enzyme of Phase II of oxidative stress response, NQO1, which follow the changes in the levels of ROS [92]. The levels of the mRNA of *NRF2*, *GCLC,* and *NQO1* in RT-expressing Ca Ski subclones did not differ from that in the Ca Ski GFP control indicating that expression of HIV-1 RT in human epithelioid cells infected with HPV16 is not associated with the induction of oxidative stress. Thus, the shift towards glycolysis in Ca Ski expressing HIV-1 RT was not associated with the change in the redox balance of the cells.

Metabolic changes often reflect changes in the mitochondrial dynamics and functionality caused by changes in the cell cytoskeleton [93,94]. We addressed this by assessing the levels of the expression in Ca Ski subclones of A-tubulin as the parameter reflecting the state of microtubules (MT) and of Y-tubulin as the parameter reflecting the condition of the MT-supported mitochondrial network [95]. Lentiviral transduction and the (over) expression of RT_A and of GFP had no effect on the expression of either *A-TUBULIN* or *Y-TUBULIN* genes which could have affected the cytoskeleton of the Ca Ski subclones and influenced their choice of metabolic pathway. Furthermore, in a previous study, we have shown that the overexpression of HIV-1 RT by murine adenocarcinoma 4T1luc2 actually restores the mitochondrial networks disrupted in the original tumor cells [72]. Overall, these findings argue against the notion that metabolic changes in Ca Ski cells could have been induced by the disruption of the cytoskeleton of Ca Ski cells.

Which properties of HIV-1 RT determine its capacity to regulate cell metabolism, apart from inducing the expression of the E6*I isoform, remains unclear. One can hypothesize that there are certain cell factors that interact with HIV-1 RT and, as a result, fail to carry out their functions in the cells causing a metabolic shift. Warren et al. described a number of cellular factors which can directly or indirectly bind to RT [96], including the kinase anchor protein 121 (AKAP121), also referred to as AKAP1 or AKAP149 (human homologue). AKAP121 is an essential regulator of the mitochondrial respiration. Decreased AKAP1 expression was detected in the glycolytic metabolism-dependent migrating cells found in invasive populations of breast cancer cells [97]. Molecular, cellular, and in silico analyses of breast cancer cell lines confirmed that AKAP1 depletion is associated with impaired mitochondrial function and dynamics, concomitant with the increased glycolytic potential and invasiveness [98]. One can hypothesize that the interaction of HIV-1 RT with AKAP1 (AKAP121) may “deplete” functional protein from the cells, causing a shift towards the glycolytic pathway.

Glucose metabolism is tightly linked to the epithelial-to-mesenchymal transition (EMT) [99]. Cancer cells employ EMT to acquire the ability to migrate, resist therapeutic agents, and escape immunity [100]. Metabolic dysregulation is known to trigger EMT [74,101,102,103,104] which, in turn, modulates cell migration [105]. The process of EMT is characterized by the profiles of the expression of a number of cellular factors, including cadherins E and N, and the transcriptional factors Twist1, Snai1 (Snail), and Snai2 (Slug). Our data show increases in the expression levels of *VIMENTIN* mRNA and a decrease in the expression level of *E-CADHERIN* in lentiviral transduction control cells expressing GFP. In an earlier study, the transfection of murine colon adenocarcinoma CT26 cells with an empty lentivirus vector (GFP-vector group) did not lead to changes in the expression of EMT factors [106]. It is likely that cell transfection does not have as strong effect as the lentiviral transduction, whereas the genomic damage caused by the transduction may lead to changes in the expression of the epithelial–mesenchymal factors. Despite this potential damage, Ca Ski RT_A subclones demonstrated an increase in the expression of *E-CADHERIN* manifesting the loss of the typical heterogeneous morphology of epithelial cells as well as an increase in the expression of mesenchymal marker *N-CADHERIN*. Ca Ski RT_A also exhibited decreased levels of the expression of *VIMENTIN*, a type III intermediate filament that is expressed in the mesenchymal cells and upregulated during cancer metastasis [100]. There were no changes in the expression of the transcription factors *TWIST*, *SNAI1* (SNAIL), and *SNAI2* (SLUG), master regulatory factors for organogenesis and wound healing tightly involved in the EMT of cancer cells [107,108,109]. These data indicated that Ca Ski RT_A subclones acquired features of the hybrid epithelial/mesenchymal (E/M) phenotype where cells simultaneously demonstrate the epithelial traits of cell-to-cell adhesion and the mesenchymal characteristics of migration and invasion [110]. Hybrid E/M cells express both epithelial markers (E-cadherin, cytokeratins, claudins, and occludins) and mesenchymal markers (N-cadherin, vimentin, and fibronectins) and may display the phenotypic characteristics of both cell types [111]. Hybrid E/M has been associated with the increased tumorigenicity of tumor cells [112], indicating the potential consequences of the presence of HIV-1 RT for Ca Ski cells.

We further assessed the effect of the biochemical and molecular characteristics of the Ca Ski subclones described above on their phenotypic features, such as doubling time, cell cycle progression, cell migration, and clonogenic activity. Lentiviral transduction and the (over) expression of RT_A or the GFP control increased cell doubling time and significantly decreased cell motility in the wound healing assay (WHA). The expression of RT_A had no effect on cell cycle progression but might have helped to overcome the G1/G0 decrease caused by lentiviral transduction (observed for the GFP control).

Lentiviral transduction in Ca Ski was found to suppress the clonogenic activity, significantly decreasing the number and size of the colonies. However, as in the case of cell migration, the (over) expression of RT_A (six RT_A inserts) reversed these effects, restoring the colony counts and the mean size of the colonies to levels characteristic of the parental Ca Ski cells. Both observations of cell motility in WHA and the results of the clonogenic analysis indicated that the overexpression of HIV-1 RT was able to compensate for the adverse effects of lentiviral transduction and/or foreign protein overexpression in Ca Ski cells.

To see the biological outcomes of the changes in in vitro cell properties, we characterized the ability of the Ca Ski RT_A subclones to form tumors in immunosuppressed mice, a well-defined property of the parental Ca Ski cells [76]. Lentiviral transduction with the expression of GFP grossly compromised the ability of Ca Ski cells to form tumors in nude mice, while the tumorigenicity of highly expressing Ca Ski RT_A was partially restored, reaching levels similar to the parental Ca Ski cells.

Taken together, the analysis of the motility/migration and clonogenic and tumorigenic activities of HPV16 infected epithelial cells in the presence of HIV-1 RT revealed that: (i) this process depends on both HIV-1 RT and HPV16 E6; (ii) although HIV-1 RT increased the expression of the E6*I isoform, the effects of HIV-1 RT and HPV16 E6 were not aligned and often were not cooperative, especially in the clonogenic activity assays with HIV-1 RT partially mitigating the effects of E6; (iii) the direct or indirect interactions of HIV-1 RT and HPV16 E6 dysregulated the cells, switching on “non-classical” traits and pathways.

## 5. Conclusions

In summary, this study demonstrates that the interplay between HIV-1 RT and HPV16 E6 modifies the phenotypic features of Ca Ski cells, affecting their motility, clonogenic activity, and tumorigenic potential, probably through the increased expression of the E6*I isoform and/or shift to glycolysis. These findings highlight the complex and sometimes opposing roles of specific molecular and cellular factors in the context of HIV-1 and HPV16 co-infection in these epithelial cells.

## 6. Limitations

Ca Ski is a cervical squamous carcinoma cell line, which is the late stage of HPV-induced carcinogenesis; thus, this model is limited in its capacity to reproduce the early stages of HPV/HIV-1 cooperation. These questions can be addressed using relevant patient derived HSIL/LSIL cells transduced with RT gene.

The level of RT present in the epithelial cells (expressed or imported) is unknown. In the Discussion section, we state that Ca Ski subclones produced an adequate amount of HIV-1 RT corresponding to the levels of protein in a natural infection (although in the epithelial cells, not CD4+ T lymphocytes). The RT protein levels could be lower in epithelial cells. This limits the power of the current model to reproduce the state of epithelial cells in an HIV-1/HPV16 co-infection.

## Figures and Tables

**Figure 1 viruses-16-00193-f001:**
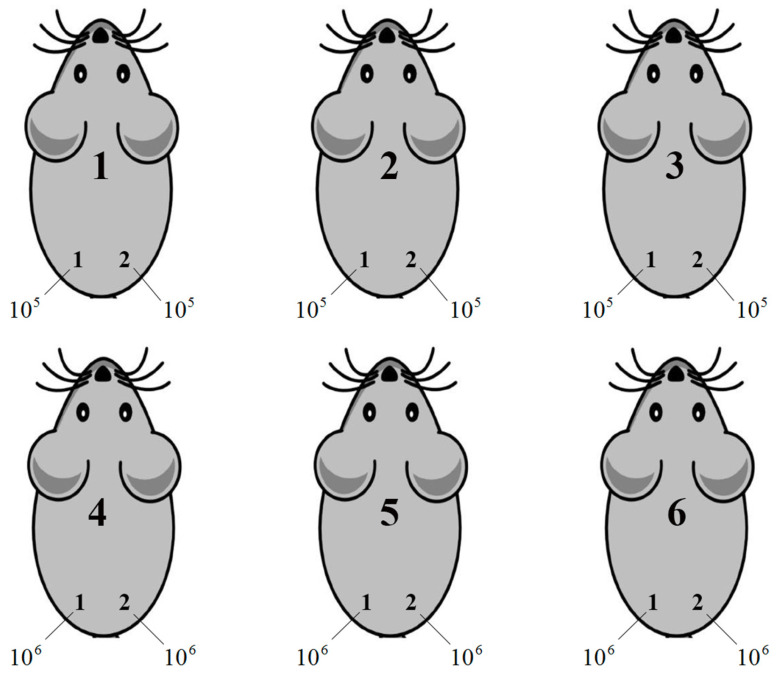
Mice were marked with a special label and placed in 4 groups of 6 animals per cage. Group 1—the control group implanted with Ca Ski; Group 2 implanted with the Ca Ski GFP G9 subclone; Group 3, mice implanted with the Ca Ski RT_A B8B5 subclone; Group 4, mice implanted with the Ca Ski RT_A H6G11 subclone. In each group, mice nn 1–3 were implanted with 10^5^ cells on the left, and 10^5^ cells on the right, and mice nn 4–6 were implanted with 10^6^ cells on the left, 10^6^ cells on the right from the back of the tail.

**Figure 2 viruses-16-00193-f002:**
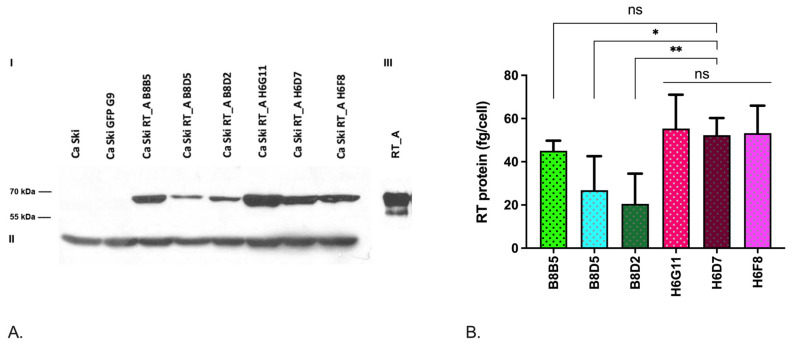
Relative expression levels of the transgene in the Ca Ski subclones with genomic inserts of the coding sequence of the consensus reverse transcriptase of HIV-1 FSU_A (RT_A). (**A**) Western blot analysis of the lysates of Ca Ski subclones stained with rabbit polyclonal anti-RT antibodies [59] (Panel I), and re-stained anti-β-actin monoclonal antibodies (Panel II).The parental Ca Ski cell line and GFP control serves as a reference. Recombinant RT_A (9 ng) resolved in PAGE after Western blotting (Panel III). The position of the weight mass markers is shown on the left. (**B**) Quantification of RT_A expression using ImageJ. The signal in the lane corresponding to a given derivative clone was quantified using the calibration curve built with the help of recombinant RT_A. The total amount of the expressed RT_A variant was divided by the number of cells used to make the lysate. The results are presented as the mean ± SD. Statistical significance was assessed using the unpaired *t* test with Bonferroni correction. ns: not significant; * *p* < 0.05; ** *p* < 0.01.

**Figure 3 viruses-16-00193-f003:**
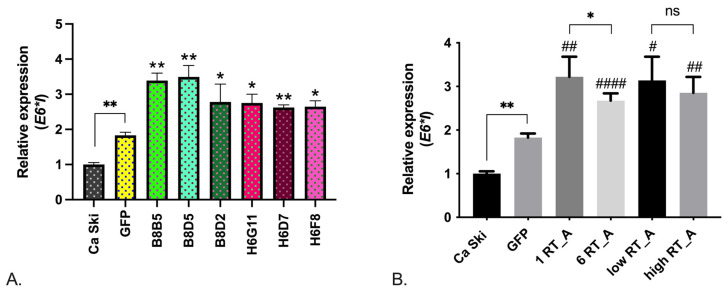
Relative mRNA expression levels of isoform I of HPV16 E6 (*E6*I*) in Ca Ski subclones expressing consensus HIV-1 FSU_A reverse transcriptase compared to Ca Ski expressing GFP and parental Ca Ski cells. (**A**) Levels of mRNA of *E6*I* in individual subclones and parental Ca Ski. * Significant difference from the value of the gene expression level in the GFP subclone (*p* < 0.05; unpaired *t* test with Bonferroni correction). (**B**) Comparison of the levels of *E6*I* mRNA in Ca Ski subclone groups according the number of RT_A DNA inserts: one RT_A (B8B5, B8D5, B8D2) and 6 RT_A (H6G11, H6D7, H6F8), and the level of RT_A expression determined by Western blot (<40 fg/cell, low, B8D5 and B8D2 subclones; >41 fg/cell, high, B8B5, H6G11, H6D7, H6F8 subclones) (Table 1). The expression level was normalized to *GAPDH* expression and calculated as a fold change compared to the parental Ca Ski line. The results are presented as the mean ± SD, ns: not significant; * *p* < 0.05; ** *p* < 0.01. # Significant difference from the value of the gene expression level in the GFP subclone (# *p* < 0.05; ## *p* < 0.01; #### *p* < 0.0001; unpaired *t* test with Bonferroni correction).

**Figure 4 viruses-16-00193-f004:**
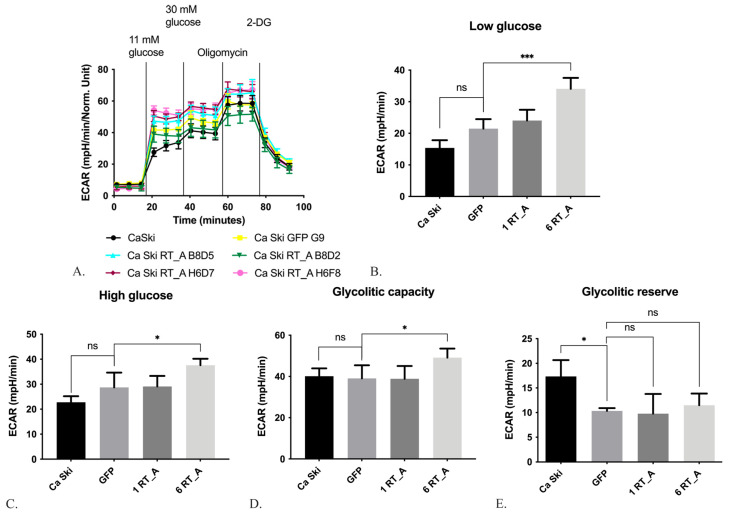
Expression of HIV-1 reverse transcriptase (RT_A) in Ca Ski cells causes an increase in the intensity of glycolysis. The glycolysis efficiency of Ca Ski derivatives was evaluated using a Seahorse analyzer according to the GlycoStress test method (**A**). The determination of ECAR in medium with a low, 11 mM (**B**), and high, 30 mM glucose concentration (**C**); maximum ECAR values upon stimulation with 1 µM oligomycin (**D**); the difference between the maximum glycolytic capacity and the level of glycolysis at 30 mM glucose (**E**). ECAR values are expressed in units of mpH/min and normalized to 1 mg of total cellular protein (mpH/min/Norm. Unit). The Ca Ski subclones with one genomic insert of the RT_A coding sequence (B8B5, B8D5, B8D2) are designated as “1 RT_A”, and those with six inserts as “6 RT_A” (H6G11, H6D7, H6F8), Ca Ski with six genomic inserts of GFP is provided as a control. The efficiency of glycolysis in individual subclones is presented in Appendix A. Histograms are presented as the mean ± SD for the analysis performed in quadruplicate. Significant difference from the value of the gene expression level in the GFP subclone (*p* < 0.05; unpaired *t* test with Bonferroni correction). ns: not significant; * *p* < 0.05; *** *p* < 0.001.

**Figure 5 viruses-16-00193-f005:**
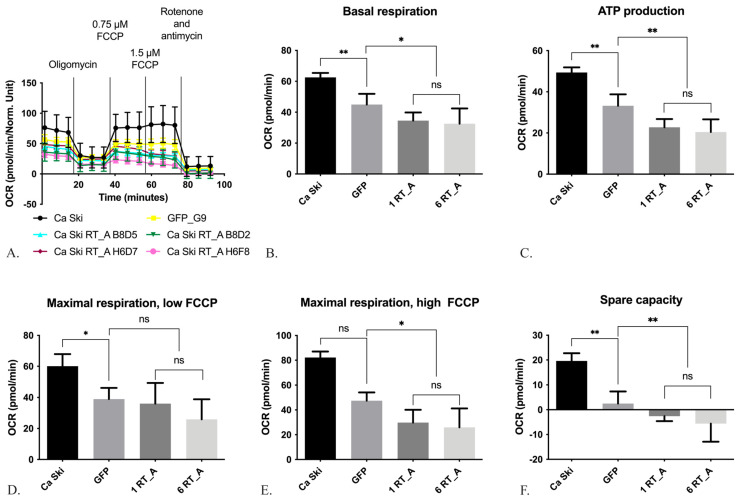
The expression of HIV-1 reverse transcriptase (RT_A) in Ca Ski cells decreases the respiratory activity of mitochondria. The respiratory activity of Ca Ski derivatives was analyzed using Seahorse technology and the MitoStress reagent kit (**A**). Basal respiration rate determined as the difference between baseline and non-mitochondrial OCR values (**B**). ATP-bound OCR determined as the difference between the basal OCR and OCR inhibited by antimycin A (**C**). Maximum OCR value stimulated by the addition of low FCCP at concentrations of 0.75 μM (**D**) and high 1.5 μM (**E**). Difference between the levels of basal and maximum respiration (**F**). OCR values are expressed in units of pmol/min and normalized to 1 mg of total cellular protein (pmol/min/Norm. Unit). Ca Ski subclones with one genomic insert of RT_A coding sequence (B8B5, B8D5, B8D2) are designated as “1 RT_A”, and those with six inserts as “6 RT_A” (H6G11, H6D7, H6F8), Ca Ski with six genomic inserts of GFP is provided as a control. The respiratory activity of mitochondria in individual subclones was presented in Appendix A. Histograms are presented as the mean ± SD for the analysis performed in quadruplicate. Significant difference from the value of the gene expression level in the GFP subclone (*p* < 0.05; unpaired *t* test with Bonferroni correction). ns: not significant; * *p* < 0.05; ** *p* < 0.01.

**Figure 6 viruses-16-00193-f006:**
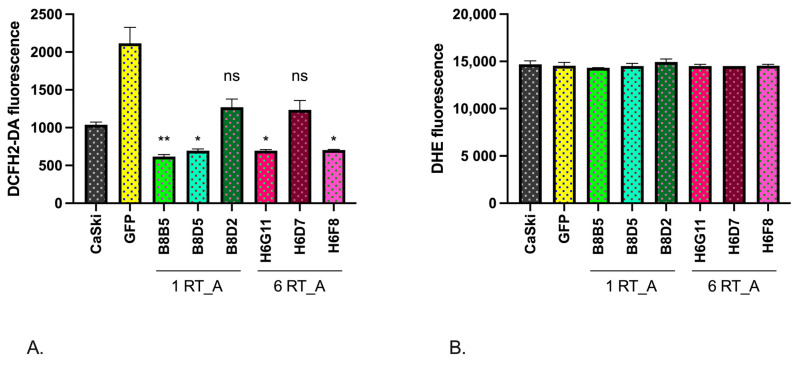
Derivatives of Ca Ski cells expressing HIV-1 FSU_A consensus reverse transcriptase variants show no ROS production. ROS production was measured using the fluorescent dyes DCFH2-DA (**A**) and DHE (**B**). Values are the mean ± standard deviation of two independent analyses performed in duplicate. ns: not significant; * *p* < 0.05; ** *p* < 0.021 using an unpaired *t* test with Bonferroni correction.

**Figure 7 viruses-16-00193-f007:**
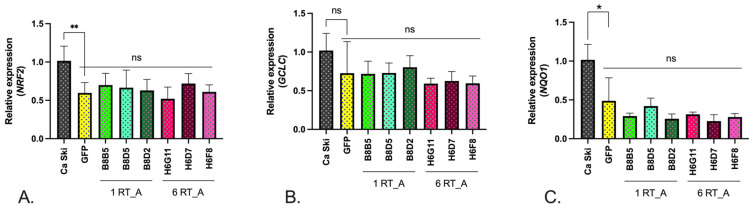
Relative expression levels of the mRNA of *NRF2* (**A**), *GCLC* (**B**), and *NQO1* (**C**) in the derivatives of Ca Ski cells expressing variants of consensus HIV-1 FSU_A reverse transcriptase. The expression levels were normalized to the expression levels of *GUSB* and calculated as a fold change compared to the parental Ca Ski line. The results are presented as the mean ± SD, ns: not significant; * *p* < 0.05; ** *p* < 0.01 by the Unpaired *t* test with Bonferroni correction.

**Figure 8 viruses-16-00193-f008:**
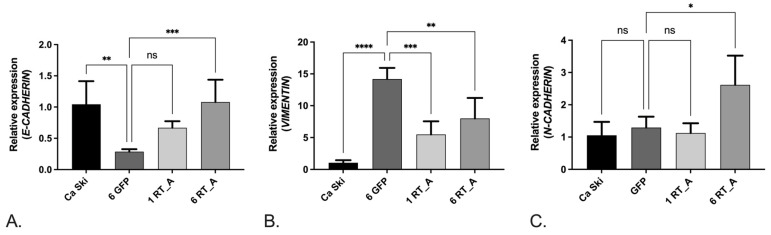
Relative mRNA expression levels of *E-CADHERIN* (**A**), *VIMENTIN* (**B**), and *N-CADHERIN* (**C**) in the derivatives of Ca Ski cells expressing HIV-1 RT_A. Ca Ski subclones with one genomic insert of RT_A coding sequence (B8B5, B8D5, B8D2) are designated as “1 RT_A”, and those with six inserts as “6 RT_A” (H6G11, H6D7, H6F8), Ca Ski with six genomic inserts of GFP are shown as a control. The expression level was normalized to *GUSB* expression and calculated as a fold change compared to the parental Ca Ski cell line. The results are presented as the mean ± SD, ns: not significant; * *p* < 0.05; ** *p* < 0.01; *** *p* < 0.001; **** *p* < 0.0001 by unpaired *t* test with Bonferroni correction.

**Figure 9 viruses-16-00193-f009:**
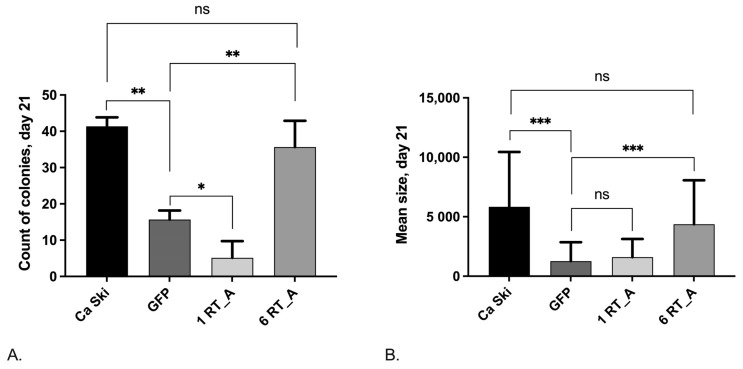
Colony counts (**A**) and the mean colony size (**B**) of Ca Ski cells, subclone with six inserts of GFP DNA (GFP) and subclones with one and six inserts of RT_A DNA (B8B5, B8D5, B8D2, and H6G11, H6D7, H6F8, respectively; Table 1) on day 21 of the assay. The evaluation of clonogenic growth in individual subclones is presented in Appendix A. Statistical significance was assessed using the Kruskal–Wallis test with Dunn’s multiple comparison test and in pairs using the Mann–Whitney test (ns: not significant; * *p* < 0.05; ** *p* < 0.01; *** *p* < 0.001).

**Figure 10 viruses-16-00193-f010:**
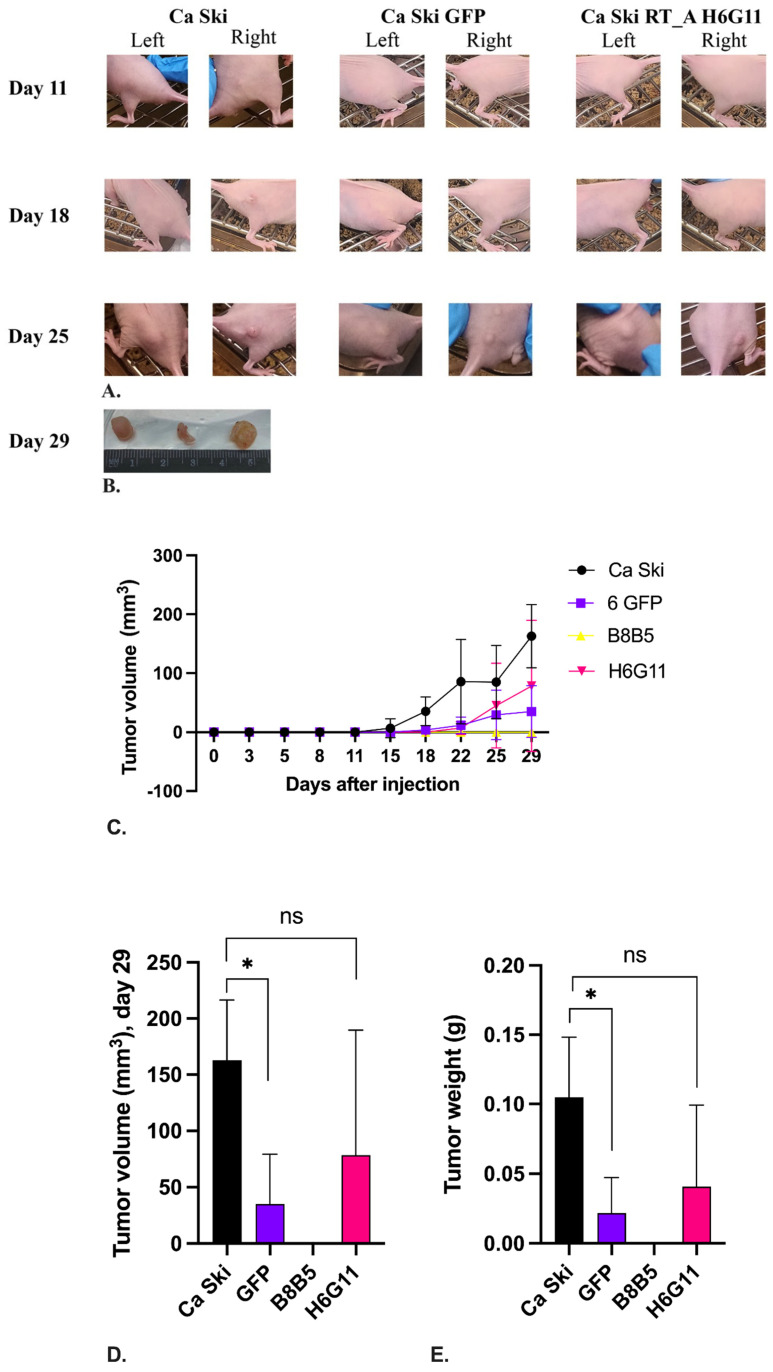
Formation of solid tumors by Ca Ski derived clones expressing HIV-1 FSU_A reverse transcriptase (RT_A) variants and EGFP fluorescent marker on days 11, 18, and 25 after ectopic implantation of 10^6^ cells in Nu/J mice (**A**). Ex vivo Ca Ski, Ca Ski GFP G9, and Ca Ski RT_A H6G11 tumor size on day 29 (**B**). General kinetics of tumor growth (**C**). Comparison of the sizes of palpable tumors on days 29 (**D**) and the weight of the tumors at the end point of the experiment (**E**). Statistical significance was assessed using the Kruskal–Wallis test with Dunn’s multiple comparison test and in pairs using the Mann–Whitney test (* *p* < 0.05).

**Table 1 viruses-16-00193-t001:** Subclones of Ca Ski expressing the consensus RT of HIV-1 clade A FSU_A strain (RT_A) obtained by lentiviral transduction of Ca Ski cells at different multiplicities of infection (MOI).

Transgene	Monoclonal Cell Line	MOI	Number of Copies per Genome	Series	Relative Expression of mRNA (*RT*), 2^−ddCt^	Quantification of RT_A Protein Expression (fg/Cell)
	B8B5	5	1		2.40 ± 0.19	45.13 ± 4.0
	B8D5	5	1	1 RT_A	1 ± 0.53	26.82 ± 13.66
RT_A	B8D2	5	1		1.99 ± 0.19	20.50 ± 12.13
	H6G11	10	6		12.61 ± 0.09	55.35 ± 12.82
	H6D7	10	6	6 RT_A	12.64 ± 0.13	52.30 ± 6.87
	H6F8	10	6		14.25 ± 0.32	53.26 ± 10.99

**Table 2 viruses-16-00193-t002:** Phenotypic characteristics of Ca Ski expressing consensus RT of HIV-1 clade A FSU_A strain (RT_A) and fluorescent marker GFP: doubling time, % of cells in the phases of the cell cycle, and the migration rate for assessed using wound healing assay (WHA).

Cells	Doubling Time, h (*n* = 12) *	% of Cells in Cell Cycle Phase (*n* = 3) **			Migration Rate in WHA, μm/h (*n* = 6) ***
		G1/G0, %	S, %	G2/M, %	
Ca Ski	25.90 ± 3.26	50 ± 3.90	24.43 ± 4.90	18.47 ± 1.25	23.84 ± 1.56
GFP	35.95 ± 7.78	38.57 ± 2.63	27.67 ± 3.62	22.3 ± 3.60	11.61 ± 2.06
B8B5	32.59 ± 6.05	45.57 ± 1.01	28.17 ± 5.92	16.67 ± 4.88	13.57 ± 1.42
B8D5	31.05 ± 7.80	50.23 ± 0.94	24.1 ± 2.68	18.17 ± 1.97	15.87 ± 0.91
B8D2	31.04 ± 7.57	47.8 ± 4.07	23.87 ± 6.53	19.53 ± 4.96	31.12 ± 3.35
H6G11	35.04 ± 7.83	41.03 ± 3.27	27.53 ± 2.87	23.03 ± 1.79	20.71 ± 2.61
H6D7	30.47 ± 6.17	43.77 ± 0.25	23.77 ± 1.15	24.27 ± 1.50	27.55 ± 1.04
H6F8	38.60 ± 9.45	44.37 ± 3.73	26.33 ± 1.59	21.93 ± 1.80	15.84 ± 1.72

*, **, ***—Number of assay repeats.

**Table 3 viruses-16-00193-t003:** Correlation of HIV-1 RT_A mRNA expression, protein production, and the expression of E6*I mRNA with the metabolic and phenotypic characteristics of Ca Ski and Ca Ski subclones (Table 1) (Spearman’s rank correlation test). Significant correlations with *p* < 0.01 (red), with *p* < 0.05 (blue); no correlation *p* > 0.05 (black).

	ECAR, Low Glucose, mpH/min	ECAR, High Glucose, mpH/min	ATP Production, pmol/min	Maximal Respiration, High FCCP, pmol/min	Relative Intensity of DCFH2-DA	mRNA N-Cadherin, 2DCt	Colony nn, Day 21	Colony Area, Day 21
RT_A mRNA, 2DCt	0.802027	0.696912	−0.798874	−0.778902	−0.13394	0.710645	0.218884	0.216436
RT_A, fg/cell	0.810437	0.764186	−0.632792	−0.587593	−0.340702	0.587969	0.003293	0.090255
mRNA E6*I, 2DCt	0.467492	0.399381	−0.603715	−0.669763	−0.619481	0.073913	−0.655339	−0.413913

## Data Availability

The data presented in this study are available on request from the corresponding author.

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
