# Peer review of "HIV-1 Reverse Transcriptase Expression in HPV16-Infected Epidermoid Carcinoma Cells Alters E6 Expression and Cellular Metabolism, and Induces a Hybrid Epithelial/Mesenchymal Cell Phenotype"

_viruses, 2024, doi:10.3390/v16020193_

Round 1
Reviewer 1 Report
Comments and Suggestions for Authors
The study by Zhitkevich et al investigated the impact of HIV-1 reverse transcriptase (RT) on cervical cancer cells infected with high-risk human papillomavirus (HPV16), a virus associated with a high incidence of epithelial malignancies in HIV-1 infection. The researchers used lentiviral transduction to express HIV-1 RT in cervical cancer cells and observed several effects. The expression of RT resulted in an increase in the E6*I isoform, suppressed mitochondrial respiration, and increased glycolysis. Additionally, the cells exhibited a hybrid epithelial/mesenchymal phenotype. Cervical cancer cells expressing RT had altered migration rate, clonogenic activity, and tumorigenic capacity compared to control cells. The study suggests that HIV-1 RT expression conferred plasticity to the cancer cells, potentially enhancing their tumorigenicity despite some negative effects of lentiviral transduction.
The study is well performed and the data was presented in a logical mannaer. I have only few minor comments.
1. Please use the full form of the abbreviated text when using first time e.g. Line 54.
2. Please check for typographic errors e.g. line 324 cell "lime" should be line
3. In all bar plots please provide individual values as dot.
4. In the table the decimal should be presented as dot not comma.
5. Please provide a limitation section.
Comments on the Quality of English LanguageNone
Author Response
Reviewer 1:
The study by Zhitkevich et al investigated the impact of HIV-1 reverse transcriptase (RT) on cervical cancer cells infected with high-risk human papillomavirus (HPV16), a virus associated with a high incidence of epithelial malignancies in HIV-1 infection. The researchers used lentiviral transduction to express HIV-1 RT in cervical cancer cells and observed several effects. The expression of RT resulted in an increase in the E6*I isoform, suppressed mitochondrial respiration, and increased glycolysis. Additionally, the cells exhibited a hybrid epithelial/mesenchymal phenotype. Cervical cancer cells expressing RT had altered migration rate, clonogenic activity, and tumorigenic capacity compared to control cells. The study suggests that HIV-1 RT expression conferred plasticity to the cancer cells, potentially enhancing their tumorigenicity despite some negative effects of lentiviral transduction.
The study is well performed and the data was presented in a logical mannaer. I have only few minor comments.
Response to Reviewer 1:
The author wish to thank Reviewer#1 for objective evaluation of the our research work and valuable comments that helped to improve the article.
- Please use the full form of the abbreviated text when using first time e.g. Line 54.
Text was corrected.
- Please check for typographic errors e.g. line 324 cell "lime" should be line
Typographic errors had been corrected.
- In all bar plots please provide individual values as dot.
We have entered individual values into the graphs, but since we had many clones, the graphs became very complicated. To read them properly, they had to be significantly increased in size, which in its turn led to significant lengthening of the manuscript. In view of these technicalities, we have chosen to introduce individual values in all Supplementary Figures, but not in the main figures of the article. Supplementary Figures are available online and can be expanded to the size convenient for viewing.
- In the table the decimal should be presented as dot not comma.
Decimals in the tables are presented as dots.
- Please provide a limitation section.
Thank you for the comment.
Ca Ski is a cervical squamous carcinoma cell line, which is the late stage of HPV-induced carcinogenesis, thus, this model is limited in its capacity to reproduce the early stages of HPV/HIV-1 cooperation. These questions can be addressed using relevant patient derived HSIL/LSIL cells transduced with RT.
The level of RT present in the epithelial cells (expressed or imported) is unknown. In discission section we state that Ca Ski subclones produced an adequate amount of HIV-1 RT corresponding to the levels of protein in natural infection (although in the epithelial cells, not CD4+ T lymphocytes). The levels of RT protein in the epithelial cells could be lower. This would increase to power of the current model to reproduce the state of epithelial cells in HIV-1/HPV16 infection.
Respective section has been added to the manuscript.

Reviewer 2 Report
Comments and Suggestions for Authors
The study by Zhitkevich et al investigates the interaction between HIV-1 reverse transcriptase (RT) and oncogenic properties of HPV16. Authors report that transfection with RT_A results in upregulation of E6*I expression, upregulation of glycolysis and suppression of mitochondrial respiration, increase in clonogenic and migratory capacity of HPV16-infected Ca Ski cells. They conclude that presence of HIV-1 RT_A improves tumorigenic capacity of Ca Ski cells in vivo. The results of the study are interesting and lay within the scope of the Viruses journal as well as the special issue on Viruses and Cellular Metabolism. However, several issues should be addressed before accepting this manuscript:
Major questions:
1. Style in the specific parts of the manuscript should be substantially improved. In particular:
a. Title of the manuscript is too long and too granular. I suggest to make it much more concise, like in authors’ previous paper, referenced in 4.
b. Avoid too long and complicated sentences (e.g., Conclusion section)
c. I suggest a stylistic and grammatic proof-reading, especially for the abstract and discussion to enhance accessibility.
2. It is still not clear for me whether effects on glycolysis on Figure 4 are associated with the copy number of RT_A in the genome or with the protein expression level? One potential clue may come from measuring glycolysis in B8B5 clone, that has a single genomic RT-A copy, but high expression of RT_A protein, although I suspect this clone is somewhat an outlier. Was B8B5 clone assessed metabolically and if yes, what is its glycolytic rate?
3. Since the authors see an increase in glycolytic rate of 6 RT_A group, I am concerned about using GAPDH gene as a reference for RT-PCR, especially in the Figure 8 and Suppl Fig 6.
4. If there is no obvious difference in doubling time, cell cycle – what causes the increase in colony formation in 6 RT_A group? Is cell death affected by the transfection with RT_A?
5. I am not convinced with the effect of 6-RT_A transfection on tumor growth in vivo, mainly due to suboptimal experimental design:
a. Key comparison should be performed between GFP-transfected and RT_A-transfected groups rather than parental cell line.
b. To comply with the ethical approval, the experiment could involve only transfected tumors without parental cell line that will allow to follow them for a longer period of time.
c. If I correctly see from Figure 1 legend, each group of mice was housed separately from other groups. According to the best animal practices, it is advisable to randomly allocate all experimental groups to the mice within the same cage. Otherwise, the observed differences in tumor growth might be a result of cage effect, rather than treatment itself.
Minor corrections:
1. Line 496: “…expression of both RT_A and GFP in Ca Ski cells caused a reduction in the production of ROS…” - cannot say that for GFP from the presented data.
2. Line 563, subtitle 3.7.1: Cannot say that RT_A increases doubling time, since GFP transfection has the same effect.
3. Line 165: The initial phrase “cell culture medium was discarded” looks out of context.
4. Mistakes/stylistic issues in lines 91, 249, 296, 662.
Comments on the Quality of English LanguageAs I said, minor proof-reading is advisable, in particular for the abstract and discussion sections
Author Response
Reviewer 2:
The study by Zhitkevich et al investigates the interaction between HIV-1 reverse transcriptase (RT) and oncogenic properties of HPV16. Authors report that transfection with RT_A results in upregulation of E6*I expression, upregulation of glycolysis and suppression of mitochondrial respiration, increase in clonogenic and migratory capacity of HPV16-infected Ca Ski cells. They conclude that presence of HIV-1 RT_A improves tumorigenic capacity of Ca Ski cells in vivo. The results of the study are interesting and lay within the scope of the Viruses journal as well as the special issue on Viruses and Cellular Metabolism. However, several issues should be addressed before accepting this manuscript.
Response to Reviewer 2:
The author wish to thank Reviewer#2 for the positive evaluation of the our study and valuable critics. Revision of the manuscript to meet these critics would has helped to significantly improve the article.
Major questions:
- Style in the specific parts of the manuscript should be substantially improved. In particular:
- Title of the manuscript is too long and too granular. I suggest to make it much more concise, like in authors’ previous paper, referenced in 4.
Current title “Expressed in HPV 16 Infected Epidermoid Carcinoma Cells, HIV-1 Reverse Transcriptase Shifts Their Metabolism towards aerobic Glycolysis, Induces Hybrid E/M Cell Phenotype and Increases Expression of E6 Transcripts” contains 28 words. Meeting the critics, we have shortened the title to 19 words: HIV-1 Reverse Transcriptase Expression in HPV16-Infected Epidermoid Carcinoma Cells alters E6 Expression, Cellular Metabolism, and induces Hybrid E/M Phenotype.
- Avoid too long and complicated sentences (e.g., Conclusion section)
Thank you very much for your comment. The text, including the conclusions, has been edited, to meet this critic.
- I suggest a stylistic and grammatic proof-reading, especially for the abstract and discussion to enhance accessibility.
Thank you very much for your comment. The abstract has been edited and shortened. The discussion has been revised and shortened. The revised text has been proof-read by the native English speakers.
- It is still not clear for me whether effects on glycolysis on Figure 4 are associated with the copy number of RT_A in the genome or with the protein expression level? One potential clue may come from measuring glycolysis in B8B5 clone, that has a single genomic RT-A copy, but high expression of RT_A protein, although I suspect this clone is somewhat an outlier. Was B8B5 clone assessed metabolically and if yes, what is its glycolytic rate?
Thank you for the comment. Unfortunately, we were limited by the capacity of the Seahorse analyzer within a single analysis. We hypothesized that changes in glycolysis are dependent on RT expression level. In the experimental outline, it was necessary to evaluate the controls: Ca Ski and GFP subclone together with the RT-expressing derivatives in each of at least three replicates, one run allowing to include only 6 samples. Due to this, we were able to assess only two cell derivatives with a high level of RT protein production (H6D7, H6F8) and two cell derivatives with a low level of RT protein production (B8D5, B8D2) at a time. Subclone B8B5 was not evaluated in this experiment. It would have served as additional evidence that -our hypothesis is correct, if B8B5 had presented similar shift to glycolysis as the 6-copy RT subclones This would illustrate that the shift was not due to introducing six lentiviral copies into the cell genome instead of one. However, we had the GFP control that had six lentiviral copies in the genome but has not demonstrated a shift to glycolysis. Subclones with 6 RT inserts demonstrated a statistically higher level of glycolysis than the GFP control, indicating a specific effect of RT. In view of this, have chosen not to repeat the experiments on B8B5 subclone.
- Since the authors see an increase in glycolytic rate of 6 RT_A group, I am concerned about using GAPDH gene as a reference for RT-PCR, especially in the Figure 8 and Suppl Fig 6.
Thank you very much for this very valuable comment. Indeed, as the protein, GAPDH has a function in the cellular metabolism, so it may not be an optimal housekeeping gene for assessing gene expression in cell lines with alterations in glycolysis. Therefore, we repeated the experiments using another housekeeping gene, GUSB. A thorough literature review did not reveal- any connection between cell metabolism/metabolic alterations, and expression of GUSB. Using GUSB, we obtained the results that had further confirmed our theory that RT induces a hybrid phenotype in the expressing cells, which is demonstrated by the increased expression of E-cadherin and N-cadherin and decreased expression of Vimentin. New results are presented in the revised Figure 8 and in the supplementary materials, (Figure S8). Once again, we wish to thank the reviewer for this valuable comment, the additional experiments strengthened our concept that HIV-1 RT induces the hybrid E/M phenotype in the affected cells.
- If there is no obvious difference in doubling time, cell cycle – what causes the increase in colony formation in 6 RT_A group? Is cell death affected by the transfection with RT_A?
Thank you for the comment. Indeed, it was shown that ferroptosis is one of the possible mechanisms of cell death in clonogenic assay [doi: 10.3389/ftox.2022.936149]. The survival in the clonogenic analysis can therefore be associated with the resistance to ferroptosis. In addition, the clonogenic activity of human osteosarcoma cell lines is improved by inhibition of N-acetylgalactosaminyltransferase enzymes family GALNT14 (which positive correlated with a key regulator of cuproptosis) [ https://doi.org/10.1155/2023/1083423]. In this study, we have shown correlation of clonogenic assay parameters with E6*I isoform expression. Specifically, expression of E6*I mRNA interfered with colony formation, reducing both the number of colonies and their size. Thus, the “toxic” factor was not HIV-1 RT, but HPV16 E6. The experiment on the clonogenic activity is consistent with the data received in the mouse model and indicates the capacity of the RT-expressing subclones with high protein expression levels to restore the clonogenic activity in vitro. The observed “restoring” or “adaptive” effect of RT contradicts the notion that RT inducing cell death in expressing clones. Moreover in our numerous investigations conducted previously on HIV-1 RT, which included transient transfection of RT gene (multiple cell lines NIH3T3, HEK293) and also constitute expression of RT (lentiviral transduction in murine adenocarcinoma cell line 4T1), we have never observed any toxicity due to RT expression, or any changes in the cell viability of RT expressing cells [DOI: 10.1016/j.vaccine.2005.08.020, DOI: 10.4161/hv.25813, DOI: 10.1155/2017/7407136]. Due to this, we have not assessed cell viability/cell death in the current study.
- I am not convinced with the effect of 6-RT_A transfection on tumor growth in vivo, mainly due to suboptimal experimental design:
- Key comparison should be performed between GFP-transfected and RT_A-transfected groups rather than parental cell line.
Thank you for the comment. GFP control subclone did not grow properly in the nude mice. These cells formed smaller tumors and not in all mice, which illustrated the negative effect of the lentiviral transduction and/or overexpression of a foreign protein on the capacity of Ca Ski cells to form tumors in nude mice. Due to incapacity of GFP subclone to grow following application of the standard protocol for implantation of Ca Ski cells into nude mice, we have chosen to use as control the parental Ca Ski cells.
- To comply with the ethical approval, the experiment could involve only transfected tumors without parental cell line that will allow to follow them for a longer period of time.
Thank you for your comment. Are initial experimental design was to use only transfected tumors (GFP and RI). However, we found that GFP subclone didn’t grow properly and could not be used as a control. Therefore, we used the parental Ca Ski cell as control which had been approved by the ethical board. Due to ethical considerations outlined in the Institutional Review Board (IRB) guidelines, we were obligated to conclude the experiment either when the first tumor in a mouse reached 1 cm3 or if the health condition of the mice deteriorated to the extent that humane sacrifice was needed. Because of these considerations we were unable to continue the experiment longer.
- If I correctly see from Figure 1 legend, each group of mice was housed separately from other groups. According to the best animal practices, it is advisable to randomly allocate all experimental groups to the mice within the same cage. Otherwise, the observed differences in tumor growth might be a result of cage effect, rather than treatment itself.
Thank you for the comment. Indeed, randomized block design involving random assignment of animals from experimental groups to cages is a useful practice that allows to address the variations in housing conditions between different cages. This practice is especially important for experiments where behavioral patterns are assessed. Randomized approach was used to divide the original mouse stocks into groups at the experimental start (after the adaptation period).
Also, in all our previous experiments, mice demonstrated similar gain of weight with time in all parallel cages. We have not observed differences in blood formula, biochemical blood parameters in mice placed in different cages within the same room and same shelf (see, for example, doi: 10.3390/microorganisms9061219). Thus, despite the fact that we did not use this practice in our study design we strongly believe that it did not compromise our results.
Minor corrections:
- Line 496: “…expression of both RT_A and GFP in Ca Ski cells caused a reduction in the production of ROS…” - cannot say that for GFP from the presented data.
Corrected.
- Line 563, subtitle 3.7.1: Cannot say that RT_A increases doubling time, since GFP transfection has the same effect.
Corrected.
- Line 165: The initial phrase “cell culture medium was discarded” looks out of context.
Corrected.
- Mistakes/stylistic issues in lines 91, 249, 296, 662.
Corrected.

Reviewer 3 Report
Comments and Suggestions for Authors
In the Manuscript (MS) by Zhitkevich et al., the authors characterized the features of HPV-16 positive CaSki cell clones transduced with one or six integrated copies of a lentiviral vector expressing Reverse Transcriptase from subtype A FSU_A strain (RT_A) or, as control, GFP. Authors demonstrated that RT_A expression leads to no changes in Oxidative Stress Metabolism and to increased expression of E6*I isoform and Glycolysis while Mitochondrial Respiration was suppressed. As the latter effects could be ascribed to cytoskeletal rearrangements, authors verified that RT_A overexpression did not lead to cytoskeletal changes. Looking at specific gene expression authors verified a change in the Epithelial Mesenchymal Transition (EMT) gene profile with altered expression of E-cadherin, SNAI1 and -2 mRNAs. Further, RT_A expression in CaSki altered cell motility as well as doubling time and cell cycle without global effects on cell cycle progression. Finally, clonogenic activity was restored in cells expressing RT_A as well as the ability of these clones to be tumorigenic when inoculated in vivo in immunosuppressed mice.
The MS by Zhitkevich et al. is well conceived and organized and the aim to investigate the effect of RT on the features of HPV positive cells cover an important topic in the field of HPV tumorigenesis in People Living With HIV-1 (PLWH). The experimental part was extensively carried out, nevertheless some main flaws limit my enthusiasm for the MS. These are my main criticisms in a point-by-point list:
1. My main concern is on the choice of the experimental procedure the authors used to express RT-A in CaSki. As they correctly stated throughout the MS the viral transduction procedure alters per se some parameters they checked (E6*I and II, N-cadherin, SNAI1 and -2 and vimentin expression, doubling time cell migration, colony formation and tumorigenic assay between the others) as evidenced by the variations recorded in the “control” clone expressing GFP. In some cases, this bias is not a confusing element (for ex. E6*I induction), but, especially in the case of “biological” assays (i.e. clonogenicity and tumorigenicity in vivo), the reduction recorded with GFP becomes a confusing bias as the level observed with RT_A clones is the same of untrasduced, parental CaSki cells. Further, GFP did not allow, as also authors stated, a proper assessment of ROS production. I’m wondering if the authors could overtake the problem due to lentiviral transduction using the plasmid they used in doi.org/10.1155/2019/6016278, simply by transfecting it into CaSki cells and by using another unrelated protein as negative control (or maybe a loss of function of RT_A mutant).
2. If you compare the expression level of RT_A by Western Blot it seems that B8B5 subclone (1 integrated copy) express more or less the same amount of RT of both H6D7 and H6F8 subclones (six integrated copies, Fig. 2A compare line 3 to lines 7 and 8). These clones should behave in a similar way, but this is not the case in some experiments the authors performed (see for ex. fig 6A, S9, S10A and B). Authors should explain these discrepancies.
3. In paragraphs 3.7.2 authors made a regression analysis “to dissect if the rate of cell migration in wound healing assay depends on any of the above studied parameters” and they stated that “Six parameters were found to predict the motility”. It is strange to me to see such a statistical analysis on an in vitro cell system rather than on a group of primary samples obtained, for ex., from PLWH and I don’t understand the predictive value of these parameters in such a system. Authors should clarify.
Minor criticisms:
Lane 296: Authors must replace “manufacturer, city, country” with the right references for these annotations about Prism9.
Author Response
Reviewer 3:
In the Manuscript (MS) by Zhitkevich et al., the authors characterized the features of HPV-16 positive CaSki cell clones transduced with one or six integrated copies of a lentiviral vector expressing Reverse Transcriptase from subtype A FSU_A strain (RT_A) or, as control, GFP. Authors demonstrated that RT_A expression leads to no changes in Oxidative Stress Metabolism and to increased expression of E6*I isoform and Glycolysis while Mitochondrial Respiration was suppressed. As the latter effects could be ascribed to cytoskeletal rearrangements, authors verified that RT_A overexpression did not lead to cytoskeletal changes. Looking at specific gene expression authors verified a change in the Epithelial Mesenchymal Transition (EMT) gene profile with altered expression of E-cadherin, SNAI1 and -2 mRNAs. Further, RT_A expression in CaSki altered cell motility as well as doubling time and cell cycle without global effects on cell cycle progression. Finally, clonogenic activity was restored in cells expressing RT_A as well as the ability of these clones to be tumorigenic when inoculated in vivo in immunosuppressed mice.
The MS by Zhitkevich et al. is well conceived and organized and the aim to investigate the effect of RT on the features of HPV positive cells cover an important topic in the field of HPV tumorigenesis in People Living With HIV-1 (PLWH). The experimental part was extensively carried out, nevertheless some main flaws limit my enthusiasm for the MS. These are my main criticisms in a point-by-point list:
Response to Reviewer 3:
The author wish to thank Reviewer#3 for objective evaluation of the our research work and valuable comments that allowed us to significantly improve the article.
- My main concern is on the choice of the experimental procedure the authors used to express RT-A in CaSki. As they correctly stated throughout the MS the viral transduction procedure alters per sesome parameters they checked (E6*I and II, N-cadherin, SNAI1 and -2 and vimentin expression, doubling time cell migration, colony formation and tumorigenic assay between the others) as evidenced by the variations recorded in the “control” clone expressing GFP. In some cases, this bias is not a confusing element (for ex. E6*I induction), but, especially in the case of “biological” assays (e. clonogenicity and tumorigenicity in vivo), the reduction recorded with GFP becomes a confusing bias as the level observed with RT_A clones is the same of untrasduced, parental CaSki cells. Further, GFP did not allow, as also authors stated, a proper assessment of ROS production. I’m wondering if the authors could overtake the problem due to lentiviral transduction using the plasmid they used in doi.org/10.1155/2019/6016278, simply by transfecting it into CaSki cells and by using another unrelated protein as negative control (or maybe a loss of function of RT_A mutant).
We have chosen GFP because it allowed a straightforward assessment of a foreign protein expression in the subclones. Negative effect of GFP in Ca Ski cells cannot be excluded (as - any other foreign protein control). Unfortunately, expressing another protein is not always as safe, and can be toxic to cells [PMID: 14555710, DOI: 10.1006/bbrc.1999.0954]. Inability to assess ROS production for GFP subclone was a limitation, but it did not affect the results of the study. By assessment of the expression of Nrf2 and of Phase II enzymes we have demonstrated that subclones were not subjected to oxidative stress.
We have chosen the transduction method because this method typically introduces foreign genes into the target cells more efficiently than plasmid transfection and provides stable levels of transgene expression over long periods of time. In contrast, transfected vector plasmids have only transient expression in cells as they do not integrate into the host genome and allow the production of a heterogeneous cell culture (some cells may contain many copies, while others carry very few or no copies). Moreover, the transient expression leads to varying protein production over time; often maximal one day post transfection and decreasing thereafter. Therefore, it is difficult to accurately estimate the level of foreign protein production in the cells (each cell) at each time point. The transient expression of the protein of interest limits all experiments that require incubation beyond 24 hours, specifically the clonogenic assay. Besides, the level of protein expression during transfection is much higher and goes beyond physiological limits. Such high expression, alongside with the process of transfection itself, can cause severe oxidative stress, which can distort the results.
Still, we understand the difficulty to assign the effects observed in the generated subclones purely RT expression. To address this, our future plan is to knockdown reverse transcriptase (RT) in RT-expressing Ca Ski subclones using CRISPR/Cas9 technology which would allow us to separate the effects of the reverse transcriptase protein from the effects of the lentiviral insert.
If you compare the expression level of RT_A by Western Blot it seems that B8B5 subclone (1 integrated copy) express more or less the same amount of RT of both H6D7 and H6F8 subclones (six integrated copies, Fig. 2A compare line 3 to lines 7 and 8). These clones should behave in a similar way, but this is not the case in some experiments the authors performed (see for ex. fig 6A, S9, S10A and B). Authors should explain these discrepancies.
We generated a panel of RT-expressing subclones to evaluate the average effect of small and larger doses of reverse transcriptase on cells. Subclones containing RT_A DNA were derived from two parental subclones B8 and H6 by limiting dilution. 1-copy subclone of B8 (MOI 5; B8B5, B8D5, B8D2) and 6-copy subclone of H6 (MOI 10; H6G11, H6D7, H6F8). The B8B5 subclone is an outlier with mutation(s)/genetic rearrangements occurring after the transduction, which could have led to the increase of expression of RT from the phosphoglycerate kinase gene (hPGK) promoter, and modified patterns of expression of other genes. Since -most of the subclones with either 1 copy (B8D5, B8D2) or 6 copy (H6G11, H6D7, H6F8) demonstrated homogenous patterns of RT expression and similar molecular properties, we chose not to dissect the reasons behind “atypical” properties of B8B5 clone. Characterizing these “atypical” properties would have required genomic sequencing and/or RNAseq analysis of this clones.
Despite differences in behavior of B8B5 as compared to other 1-copy clones, we have chosen to keep the descriptive data for this clone in the study. Interestingly, this has not led to a radical change in the outcome of data analysis.
- In paragraphs 3.7.2 authors made a regression analysis “to dissect if the rate of cell migration in wound healing assay depends on any of the above studied parameters” and they stated that “Six parameters were found to predict the motility”. It is strange to me to see such a statistical analysis on an in vitrocell system rather than on a group of primary samples obtained, for ex., from PLWH and I don’t understand the predictive value of these parameters in such a system. Authors should clarify.
Thank you very much for this valuable comment. We discussed your suggestion with our colleagues and co-authors and decided to delete this section from the manuscript. This has made the manuscript easier to read and more compact, which was recommended by the other Reviewers.
Minor criticisms:
Lane 296: Authors must replace “manufacturer, city, country” with the right references for these annotations about Prism9.
Corrected.

Round 2
Reviewer 3 Report
Comments and Suggestions for Authors
Even if the Authors answered my methodological criticisms, the limitations of this study stay in place. Nevertheless, the Authors stated them in the new “Limitations” section, then warning the readers about these limitations.
Only a minor revision should be still applied: the sentence corresponding to lines 927-931 refers to results obtained by the correlation analysis that Authors removed, then it must be removed too.
Only a minor revision should be still applied: the sentence corresponding to lines 927-931 refers to results obtained by the correlation analysis that Authors removed, then it must be removed too.
Author Response
Dear Editor,
thank you very much for organizing the review process of the manuscript 2811363 by Alla Zhitkevich et al “HIV-1 Reverse Transcriptase Expression in HPV16-Infected Epidermoid Carcinoma Cells alters E6 Expression, Cellular Metabolism, and induces Hybrid E/M Cell Phenotype”.
We have carefully addressed the comments of the Reviewer and introduced respective changes in the manuscript in tracking mode. Our response to the Reviewer comments is given below.
Also, one of the co-authors suggested changing the abbreviation E/M in the title of the manuscript to the full version: Epithelial/Mesenchymal, so we would like to request a title change to the following version: “HIV-1 Reverse Transcriptase Expression in HPV16-Infected Epidermoid Carcinoma Cells Alters E6 Expression and Cellular Metabolism, and Induces a Hybrid Epithelial/Mesenchymal Cell Phenotype ".
We hope that you will find the revised manuscript acceptable for publication in Viruses.
Alla Zhitkevich and Maria Isaguliants,
acting on behalf of all authors.
Reviewer 3:
Even if the Authors answered my methodological criticisms, the limitations of this study stay in place. Nevertheless, the Authors stated them in the new “Limitations” section, then warning the readers about these limitations.
Only a minor revision should be still applied: the sentence corresponding to lines 927-931 refers to results obtained by the correlation analysis that Authors removed, then it must be removed too.
Only a minor revision should be still applied: the sentence corresponding to lines 927-931 refers to results obtained by the correlation analysis that Authors removed, then it must be removed too.
Response to Reviewer 3:
The author wish to thank Reviewer#3 for valuable comments that allowed us to significantly improve the article.
Thank you very much for the comment. Only multiple regression analyzes were removed from the manuscript, but not correlation analyses. However, indeed, you correctly noted that no statistical analysis was presented to derive the conclusion presented in the corresponding sentence. Therefore, we added it to the supplementary materials in the form of Table S2.